# Characterizing optogenetically mediated rebound effects in anaesthetized mouse primary visual cortex

Jared T. Shapiro, Nicole M. Michaud and Nathan A. Crowder 

*Department of Psychology and Neuroscience, Dalhousie University, Halifax, Nova Scotia, Canada*

Handling Editors: Nathan Schoppa & Conny Kopp-Scheinpflug

The peer review history is available in the Supporting Information section of this article (https://doi.org/10.1113/JP287265#support-information-section).

**Abstract figure legend** Many studies of cortical circuits use optogenetic activation of inhibitory interneurons to suppress pyramidal cell activity, but after the light is turned off pyramidal cells sometimes show excess spiking, which is called a post-inhibitory rebound. We recorded neurons in mouse visual cortex to test whether optogenetically mediated post-inhibitory rebounds were affected by the activity and membership of the local cortical network. Visual stimuli that strongly drive visual cortex activity increased the magnitude of both post-inhibitory rebounds in pyramidal cells and novel post-excitation rapid decay (PERD) within the directly optogenetically activated interneurons. We also found that activation of parvalbumin-expressing interneurons generally produced larger rebound effects than other interneuron subtypes. These optogenetically mediated rebound effects provide insights into the cellular and network mechanisms that regulate excitation and inhibition in the cortex.

**Jared T. Shapiro** completed his Bachelor's and Master's degrees in Neuroscience at Dalhousie University in Halifax, Nova Scotia, where his research focused on cortical circuits in mouse visual cortex. Since graduating, he has been working in the Izaak Walton Killam Health Centre paediatric emergency department. His neuroscience research and frontline clinical work continue to deepen his interest in physiology, but he is now focused on its application to patient care and medical discovery.

**Abstract** Optogenetic tools have been used to investigate neural circuits in mouse primary visual cortex (V1), where channelrhodopsin-mediated activation (photostimulation) of inhibitory interneuron subtypes expressing parvalbumin (Pvalb+), somatostatin (SOM+) or vasoactive intestinal peptide (VIP+) can alter the responses of excitatory pyramidal neurons. Some studies have mentioned rebound spiking after this photostimulation, but no systematic analysis of these post-inhibitory rebound effects has yet been performed. Here, we characterized optogenetically mediated rebound effects in pyramidal cells and interneurons following Pvalb+, SOM+ or VIP+ photostimulation in isoflurane anaesthetized mice and investigated whether V1 network features such as activity and connectivity can affect rebound magnitude. We found converging evidence that rebounds were largest when interneuron photostimulation was coupled with visual stimuli that strongly activate V1. Many directly photostimulated interneurons showed post-activation effects that differed from rebounds in polarity and timing. Finally, Pvalb+ photostimulation produced the largest rebounds. Our findings suggest that both cellular and network mechanisms contribute to rebound effects in mouse V1.

(Received 13 July 2024; accepted after revision 24 June 2025; first published online 11 July 2025)

**Corresponding author** N. A. Crowder: Department of Psychology and Neuroscience, Dalhousie University, 1355 Oxford Street, PO Box 15000, Halifax, Nova Scotia, Canada, B3H 4R2.    Email: nathan.crowder@dal.ca

## Key points

- To study cortical circuits, light-activated optogenetic proteins targeted to inhibitory interneurons are used to suppress excitatory pyramidal cells, but after the light is turned off pyramidal cells sometimes show excess spiking, which is called a post-inhibitory rebound.
- We investigated whether optogenetically mediated post-inhibitory rebounds are affected by local cortical network activity and connectivity in anaesthetized mouse visual cortex.
- We show that visual stimuli that strongly activate visual cortex increase the magnitude of both post-inhibitory rebounds in pyramidal cells and novel post-excitation effects in the directly optogenetically activated interneurons.
- Activating different interneuron subtypes, each with distinct connection patterns within the local network, elicits different rebound effects.
- The properties of optogenetically mediated rebound effects in cortex can provide insights into how excitation and inhibition are regulated during normal brain function.

## Introduction

A fundamental aim of systems neuroscience is to determine how neural circuits process information. The mouse primary visual cortex (V1) has become a critical brain area for studying cortical circuits, where light-sensitive optogenetic proteins have been used to modulate the activity of GABAergic interneurons to investigate how they contribute to specific receptive field properties (Wood et al., 2017). *In vitro* studies have described a V1 circuit constructed with excitatory pyramidal cells and several distinct GABAergic interneuron subtypes segregated into three families based on parvalbumin (Pvalb+), somatostatin (SOM+) and vasoactive intestinal peptide (VIP+) expression, which differ in morphology, functional connectivity and distribution across cortical layers (Rudy et al., 2011; Tremblay et al.,

2016). Many *in vivo* investigations into the potential functions of Pvalb+, SOM+ or VIP+ cells in V1 have focused on how the visual response properties of pyramidal cells can be altered while a particular interneuron subtype is being optogenetically photostimulated (Atallah et al., 2012, 2014; El-Boustani & Sur, 2014; Lee et al., 2012, 2014; Shapiro, Michaud, et al., 2022; Wilson et al., 2012; Wood et al., 2017). However, this optogenetic modulation can also elicit excess spikes after light offset compared to control conditions, producing a so-called 'rebound' of neuronal activity during the post-photostimulation epoch (Chang et al., 2018; Chuong et al., 2014; Edgerton & Jaeger, 2014; Lado et al., 2022; Li et al., 2019; Madisen et al., 2012; Mahn et al., 2016; Sessolo et al., 2015; Stark et al., 2013; Takeuchi et al., 2016; Tønnesen et al., 2009; Tsunematsu et al., 2011; Witter et al., 2013). The occurrence of

rebound effects has been briefly noted in studies of V1 (Ayzenshtat et al., 2016; Chuong et al., 2014) and higher visual areas (Jin & Glickfeld, 2020), but they have not been extensively investigated. Therefore, we re-analysed archival data from previously published experiments (Shapiro, Gosselin, et al., 2022; Shapiro, Michaud, et al., 2022), and collected new electrophysiological data, to characterize optogenetically mediated rebound effects within the active neural circuitry of V1, thereby gaining insights into the functioning of cortical circuitry.

Before the advent of optogenetics, the field of cellular neuroscience studied post-inhibitory rebounds because of the critical role rebound spikes are proposed to serve in oscillatory circuits or central pattern generators (Huguenard & McCormick, 2007; McCormick & Bal, 1997; Wahl-Schott & Biel, 2009), which have been implicated in sustaining biological rhythms, sensory processing and behaviours such as spatial navigation (Bal et al., 1995; Bucher et al., 2015; Felix et al., 2011; Ferrante et al., 2017; Huguenard & McCormick, 2007; Kulesza et al., 2007; Marder & Bucher, 2001; Marder & Calabrese, 1996; McCormick & Bal, 1997; Rajaram et al., 2019; Satterlie, 1985). Much of this work was conducted *in vitro* in subcortical brain areas to identify the underlying ionic currents that produce post-inhibitory rebound spikes, with the best studied being hyperpolarization-activated ($I_h$) currents mediated by hyperpolarization-activated cyclic-nucleotide-gated (HCN) channels and low-threshold $Ca^{2+}$ currents (Aizenman & Linden, 1999; Angstadt et al., 2005; Ascoli et al., 2010; Barrio et al., 1994; Bertrand & Cazalets, 1998; Biel et al., 2009; Felix et al., 2011; Ferrante et al., 2017; Sun & Wu, 2008; Sun et al., 2020; Zheng & Raman, 2011). When a neuron is strongly hyperpolarized (e.g. by current injection), this activates $I_h$ currents that depolarize the neuron, which subsequently activates low-threshold $Ca^{2+}$ currents to generate rebound depolarization and spikes (Benarroch, 2013; McCormick & Bal, 1997; Perez-Reyes, 2003; Wahl-Schott & Biel, 2009). The magnitude of hyperpolarization seems to affect post-inhibitory rebound spiking probability and latency (Tadayonnejad et al., 2009). Importantly, there are relatively few neurons where rhythmic firing (and presumably rebound spiking) is completely mediated by $I_h$ (Bean, 2025). Additional mechanisms for generating rebounds that are not mutually exclusive to $I_h$ could include the deinactivation of inward $Na^+$ currents, persistent $Na^+$ currents and deactivation of slow outward currents (Angstadt et al., 2005; Calabrese & Feldman 1999; Franceschetti et al., 1995; Taddese & Bean, 2002).

Optogenetically mediated rebounds can arise following two main circumstances that produce inhibition. First, inhibitory optogenetic proteins (e.g. halorhodopsin, archaerhodopsin) producing direct hyperpolarization by light have been shown to trigger rebound spikes after light offset when they are expressed in cortical pyramidal cells (Chuong et al., 2014; Madisen et al., 2012; Tønnesen et al., 2009), interneurons (Ayzenshtat et al., 2016; Lado et al., 2022), thalamocortical projections within the cortex (Mahn et al., 2016), tyrosine-hydroxylase-expressing neurons in the locus coeruleus (Takeuchi et al., 2016) and orexin-expressing neurons in the lateral hypothalamic area (Tsunematsu et al., 2011). Second, when interneurons express excitatory optogenetic proteins (e.g. channelrhodopsin), activating these interneurons sends inhibition to post-synaptic neurons, which can elicit rebound spikes in the post-synaptic cells after light offset (Chang et al., 2018; Jin & Glickfeld, 2020; Li et al., 2019; Sessolo et al., 2015; Stark et al., 2013). After light offset, previously suppressed neurons can show higher resting membrane potentials than prior to light onset (Mattis et al., 2012), which may make it easier for these cells to fire post-photostimulation spikes. Some optogenetically mediated rebounds can be produced by the classic $I_h$ mechanism. For example, pharmacological blockade of HCN channels in hippocampus abolished optogenetically mediated rebounds (Stark et al., 2013). However, there are also observations that indicate alternative mechanisms could produce rebounds in other brain regions or cell types. Notably, optogenetically mediated rebounds in somatosensory cortex are unaffected by HCN blockade but abolished by tetrodotoxin (TTX; Chang et al., 2018). Furthermore, several types of anaesthetics reduce $I_h$ (Goldstein, 2015), but robust optogenetically mediated rebounds have been measured under isoflurane anaesthesia (Chuong et al., 2014). Despite these insights, optogenetically mediated rebound effects have generally been considered a nuisance, prompting the use of ramp-like light offsets to avoid rebound spiking in cortical pyramidal cells or thalamocortical projections in the cortex (Chuong et al., 2014; Mahn et al., 2016), and so deliberate work rigorously examining the properties of rebounds with optogenetics is scant.

With its well-characterized neural circuitry, V1 is an ideal testing ground to explore how cellular and network mechanisms work together to produce optogenetically mediated rebounds. In the present study, we examined the effects of Pvalb+, SOM+ or VIP+ photostimulation on V1 activity in isoflurane anaesthetized mice to investigate two network features that are likely to impact rebound magnitude: activity and membership. We first manipulated local network activity within V1 by presenting visual stimuli that produced strong or weak afferent drive due to specific feed-forward connections from thalamus (Hubel & Wiesel, 1962; Movshon et al., 1978; Ohzawa et al., 1985; Priebe & Ferster, 2008). Therefore, rebounds elicited by a particular level of interneuron photostimulation were measured against the

backdrop of different levels of ongoing network activity. If optogenetically mediated rebounds in pyramidal cells are primarily produced by hyperpolarization that activates $I_h$ currents (Stark et al., 2013; Wahl-Schott & Biel, 2009), then visual stimuli that produce excitation should counteract this hyperpolarization and reduce rebounds. Alternatively, rebounds could actually be larger with visual stimuli that strongly drive pyramidal cells if they are produced by other cellular mechanisms such as persistent $Na^+$ currents (Bean 2025; Taddese & Bean, 2002), or network mechanisms such as release from inhibition (Singer, 1996). Second, we explored how rebounds might be mediated by unique members within a network by photostimulating each of the three different interneuron subtypes in V1. These subtypes differ in their connectivity, forming a well-characterized neural circuit: Pvalb+ cells inhibit all neuron subtypes including themselves; SOM+ cells inhibit all neuron subtypes excluding themselves; and VIP+ cells mainly inhibit SOM+ cells but can also inhibit or excite each other weakly (Fig. 1*A*; Karnani et al., 2016; Pfeffer et al., 2013; Tremblay et al., 2016). There are also differences between interneuron subtypes in the thalamocortical input they receive (Ji et al., 2016), how they summate excitatory inputs (Kapfer et al., 2007; Silberberg, 2008; Silberberg & Markram, 2007) and regions of pyramidal cells their synapses target (Fig. 1*A*; Pfeffer et al., 2013; Tremblay et al., 2016). Therefore, we reasoned that photostimulating each interneuron subtype separately may produce distinct rebound effects. We were surprised to find that photostimulating each interneuron subtype could produce both post-excitatory suppression within the directly activated interneurons themselves and post-inhibitory rebound spiking in the pyramidal cells receiving inhibition from them. Rebound size and prevalence were highly variable across all datasets, but overall Pvalb+ photostimulation produced the largest rebounds. Most importantly, we found converging evidence from multiple experiments that coupling interneuron photostimulation with visual stimuli that produced higher levels of afferent drive to V1 neurons elicited larger rebounds in more cells.

## Methods

### Ethical approval

All experiments were conducted in accordance with the guidelines of the Canadian Council on Animal Care and were approved by the Dalhousie University Committee on Laboratory Animals (protocol numbers: 16-078, 18-084). The authors confirm that they understand the ethical principles under which the *Journal of Physiology* operates and that their work complies with the editorial by Grundy (2015).

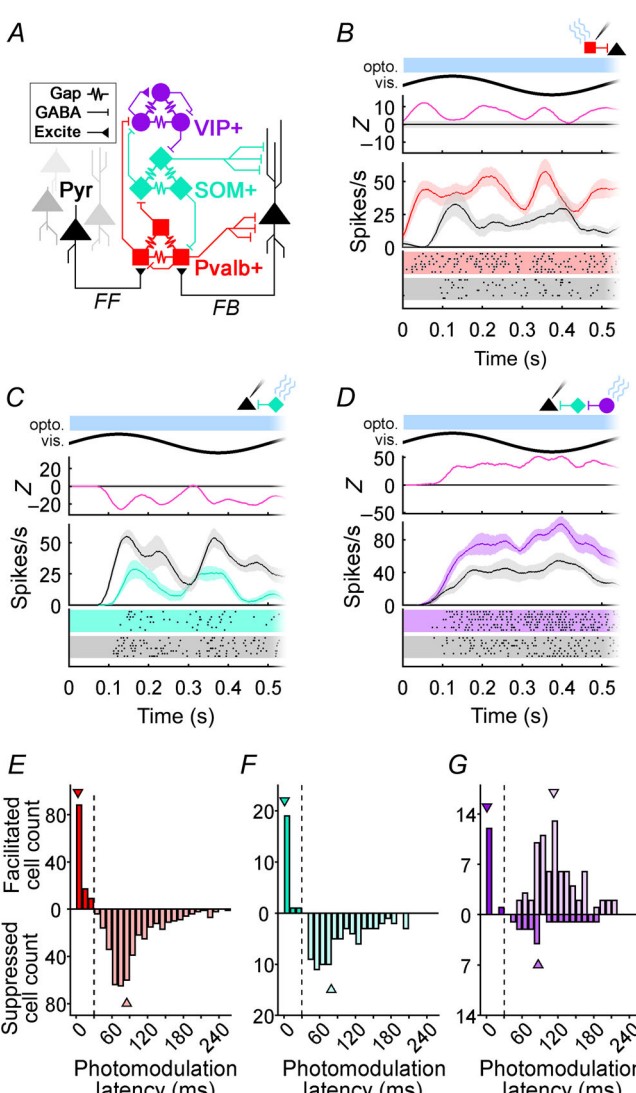

**Figure 1. Optogenetic photomodulation latencies of mouse V1 neurons**
*A*, wiring diagram of mouse V1 circuitry outlined *in vitro* (Karnani et al., 2016; Pfeffer et al., 2013). Pvalb+ (red squares), SOM+ (teal diamonds) and VIP+ (purple circles) interneurons are connected with electrical synapses (Gap) as well as GABAergic (GABA) and cholinergic chemical synapses. Pyr cells (black triangles) use excitatory glutamatergic chemical synapses (Excite) to activate interneurons with feedforward (FF) or feedback (FB) connections. *B*, spike density function (SDF) and raster plot showing a Pvalb+ interneuron's response to a drifting 100% contrast sine wave grating without (black line) or with optogenetic activation (red line). Shading around SDFs indicates SEM. The *Z*-score function above the SDF indicates the difference in firing between optogenetic and control conditions (pink line). Grey area surrounding the zero-line indicates *Z*-scores of ±2, which was the threshold to determine a neuron's photomodulation latency. The timing of photostimulation (opto.) and the visual stimulus (vis.) are indicated at the top of the graph by the blue bar and bold black sinewave, respectively. *C* and *D*, SDFs and *Z*-scores showing the responses of Pyr cells during SOM+ and VIP+ photostimulation, respectively. The format is identical to *B*, except the signature colour for each interneuron subtype is taken from *A*. *E*–*G*, histograms plotting photomodulation latencies from

neurons recorded in Pvalb-Ai32 (red; $n = 540$), Som-Ai32 (teal; $n = 101$) and Vip-Ai32 mice (purple; $n = 112$). The vertical dotted lines indicate the 30 ms threshold used to segregate photostimulated interneurons from post-synaptic putative pyramidal cells (see Results). Arrowheads indicate the median photomodulation latencies for interneuron or Pyr groups from each mouse line.

## Animals

Transgenic mice for optogenetic experiments were produced by crossbreeding Ai32 (JAX stock # 012569) mice with Pvalb-IRES-Cre (JAX stock # 008069), Sst-IRES- Cre (Jax Stock #013044) or Vip-IRES-Cre (Jax Stock #010908) mice from The Jackson Laboratory (Bar Harbor, ME, USA). Transgenic mice offspring exclusively expressed Channelrhodopsin 2 (ChR2[H134R]-EYFP) in Pvalb+ (Pvalb-Ai32 mice), SOM+ (Som-Ai32 mice) or VIP+ cells (Vip-Ai32 mice), which allowed for precise photostimulation of these interneuron subtypes *in vivo*. We ensured crossbreeding accuracy by genotyping with standard PCR testing. Part of this study involved a novel reanalysis focusing on the post-photostimulation epoch in electrophysiological recordings that were also used in previous publications (Shapiro, Gosselin, et al., 2022; Shapiro, Michaud, et al., 2022), which were obtained from a total of 31 (16 female) Pvalb-Ai32, 13 (five female) Som-Ai32 and eight (five female) Vip-Ai32 transgenic mice. Novel data for the present study were collected from 11 (six female) Pvalb-Ai32 mice. All mice were 2–9 months old and weighed 20–30 g. Electrophysiological recordings were pooled across mice, so *n* values reported in the Results always refer to the number of neurons. Experimenters were not blinded to the genotype of the mice. Mice were housed in groups of two or three in standard plastic laboratory mouse cages, containing cellulose bedding, small tubes for enrichment, nesting material, and a hopper with *ad libitum* access to water and standard rodent chow. Colony rooms followed a reverse light–dark cycle (12 h: lights off from 09.30 to 21.30 h).

## *In vivo* electrophysiology

Mice were sedated with chlorprothixene (Sigma Aldrich, St Louis, MO, USA; 5 mg/kg, I.P.) then anaesthetized with isoflurane in oxygen (∼2.5% isoflurane during induction, ∼1.5% during surgery and ∼0.5% during recording; Pharmaceutical Partners of Canada). Chlorprothixene top-up doses were administered every 4 h. Body temperature was maintained at 37.5°C with a heating pad. The scalp was retracted, and the skull was immobilized with a head-post using dental epoxy. A small craniotomy (∼1 mm$^2$) over V1 (0.8 mm anterior and 2.3 mm lateral from lambda; Paxinos & Franklin, 2019) was made for electrophysiological recordings and optogenetic photostimulation. Petroleum jelly formed a well around the craniotomy, which was filled with saline to prevent dehydration of the cortex. We protected the corneas with frequent application of optically neutral silicone oil (30,000cSt, Sigma Aldrich). Mouse eye movements under anaesthesia have been shown to be negligible (Gao et al., 2010; Niell & Stryker, 2008; Wang & Burkhalter, 2007), so eyes were not immobilized, and pupils were not dilated to maintain a large depth of focus. For preparatory surgery, and any subsequent minor procedures that might cause pain, the anaesthetic depth was tested with the toe-pinch reflex, and procedures were only initiated once mice reached the anaesthetic surgical plane. During recording, we monitored breath rate and muscle tone to ensure a steady depth of anaesthesia was maintained. Additional analgesics were not administered because animals were anaesthetized throughout the experiments and killed without ever gaining consciousness, and no neuro-muscular blocking agents were used. At the conclusion of electrophysiological recordings isoflurane anaesthesia was increased to 2.5% and mice were killed with cervical dislocation.

Initial extracellular recordings were performed using a glass micropipette (tip diameter of 2–5 μm filled with 2 M NaCl) to localize the monocular retinotopic portions of V1 (∼30–90° azimuth and 10–40° elevation), which was then replaced with a tetrode for multi-channel recordings (Teflon-coated NiCr wire with gold-plated tips; impedance ∼300 kΩ). Electrophysiological signals were bandpass filtered (50–2000 Hz) and sampled (25 kHz) with a CED 1401 digitizer and Spike2 software (Cambridge Electronic Design, Cambridge, UK). Online analyses were performed in Spike2 from triggered transistor–transistor logic (TTL) pulses from a window discriminator (WD-2, Dagan, Minneapolis, MN, USA). Offline spike sorting was conducted with Spike2 software using a supervised template-matching algorithm. Single units were identified in multi-unit recordings by generating templates concatenated from the four tetrode channels to produce unique spike-waveforms for each unit. We ensured only well-isolated units were included in our analysis by selecting only those units that had clearly separated clusters from principal component analysis and inter-spike interval histograms with >1% of spikes separated by <1 ms.

## Optogenetic photostimulation

For all experiments we optogenetically activated either Pvalb+, SOM+ or VIP+ interneurons using a 470 nm fibre-coupled LED (0.4 mm diameter; 0.39 NA; Thorlabs, Newton, NJ, USA) situated ∼0.2–0.5 mm above the exposed V1 surface. The CED 1401 synchronized the timing of LED activation and visual stimulus presentation. Our LED power output was sufficiently strong

to silence V1 (0.002–2.1 mW; Atallah et al., 2012; King et al., 2016), and therefore at each recording site we used online analysis to select a light intensity that produced moderate suppression for the experimental protocols (0.06–0.44 mW/mm$^2$ at the cortical surface; median = 0.18 mW/mm$^2$). This dim cortical illumination was applied in a continuous block to maintain potentially important temporal features of neural firing and rebounds (e.g. onset transients, firing rate decay, phase preference). Finally, at the intensities used, 470 nm light diffuses a few hundred micrometres laterally from the fibreoptic (Stujenske et al., 2015), though activation travelling through gap junction connected networks of interneurons can reach a radius of ∼1 mm (Li et al., 2019), which encompassed the area of V1 expected to be retinotopically activated by the visual stimuli used in this study (Marshel et al., 2011).

## Visual stimulus

Visual stimuli were programmed in MATLAB using the Psychophysics Toolbox extension (Brainard, 1997; Pelli, 1997), and presented on a calibrated CRT monitor (LG Flatron 915FT Plus 19 inch display, 100 Hz refresh, 1024 × 768 pixels, mean luminance = 30 cd/m$^2$) at a viewing distance of 15–30 cm. Flashed vertically oriented bars used in Experiment 2 subtended the full screen height. Drifting sine-wave gratings used in experiments 1 and 3 were presented in a circular aperture surrounded by a grey field of mean luminance. All gratings had a spatial frequency (SF) of 0.03 cycles per degree and temporal frequency (TF) of 2 Hz, which approximated mouse V1 neuron population averages (LeDue et al., 2012; Niell & Stryker, 2008). These gratings elicited robust responding from many V1 neurons within multi-unit recordings due to broad spatiotemporal tuning of the neurons across several octaves of SF and TF (King et al., 2015; LeDue et al., 2012; LeDue et al., 2013; Niell & Stryker, 2008). At each recording site, we first performed preliminary analyses of receptive field (RF) size (4–64° diameter apertures) and orientation/direction (22.5° resolution) preferences by constructing online tuning functions using TTL triggering of the best isolated unit, which aided in selecting appropriate stimulus parameters for ensuing experimental recordings.

**Experiment 1: contrast effects.** In Experiment 1 we examined rebounds produced when a set level of Pvalb+, SOM+ or VIP+ photostimulation was paired with different levels of afferent drive to V1 from gratings of varying contrasts (reanalysis of Shapiro, Michaud, et al., 2022). Grating contrast was defined as:

$$\% \ Michelson \ Contrast = \frac{(L_{max} - L_{min})}{(L_{max} + L_{min})} \times 100 \quad (1)$$

where $L_{max}$ and $L_{min}$ are the maximum and minimum luminance in the grating, respectively. Five grating contrast levels (6, 12, 24, 50 and 100%) were presented for 1 s each in random order followed by a homogeneous grey screen lasting 2 s during the interstimulus interval (ISI). Photostimulated trials were randomly interleaved with non-photostimulated control trials. Photostimulation lasted for 1 s and completely overlapped in time with the visual stimuli (Figs 2 and 5A), which produced photostimulation and post-photostimulation epochs to examine optogenetic modulation and rebound effects, respectively. We recorded 8–12 repetitions of each contrast level for both control and photostimulation trials.

**Experiment 2: spatial effects.** In Experiment 2 we examined rebounds produced when a set level of Pvalb+ photostimulation was paired with stationary bar stimuli flashed inside or outside of the RF to elicit varying levels of afferent drive (reanalysis of Shapiro, Gosselin, et al., 2022). During each trial, a single black or white vertically oriented bar was flashed in one of nine randomly selected locations over a homogeneous grey background of mean luminance (Fig. 7B). The bar appeared at 100 ms into each trial then vanished at 600 ms, and the grey background remained on the monitor for an additional 400 ms before the next trial was initiated. On photostimulated trials, the LED was on for the first 600 ms, which produced photostimulation and post-photostimulation epochs to examine optogenetic modulation and rebound effects, respectively. Control and photostimulation trials at all nine bar locations were randomly interleaved, and each condition was presented for 20 repetitions.

**Experiment 3: timing effects.** In Experiment 3 we examined whether rebound magnitude was influenced by afferent drive to V1 during either the pre-photostimulation, photostimulation or post-photostimulation epochs by varying the timing of Pvalb+ photostimulation relative to the appearance of a drifting grating. The visual stimulus in all conditions was a 100% contrast drifting grating that appeared 500 ms into the trial and disappeared at 1500 ms. During photostimulated trials, the LED was turned on for 500 ms at four different delays to create four distinct conditions named after the level of afferent drive during the photostimulation and post-photostimulation epochs (Fig. 8A). In the 'Lo-Hi' condition, the cortex was illuminated in the first 500 ms before the grating appeared, and thus there was low afferent drive when the LED was on, but high afferent drive in the post-photostimulation epoch due to the appearance of the grating (Fig. 8A bottom-right). In the 'Hi-Hi' condition, the cortex was illuminated between 501 and 1000 ms, and thus there was high afferent drive when the LED was on due to the appearance

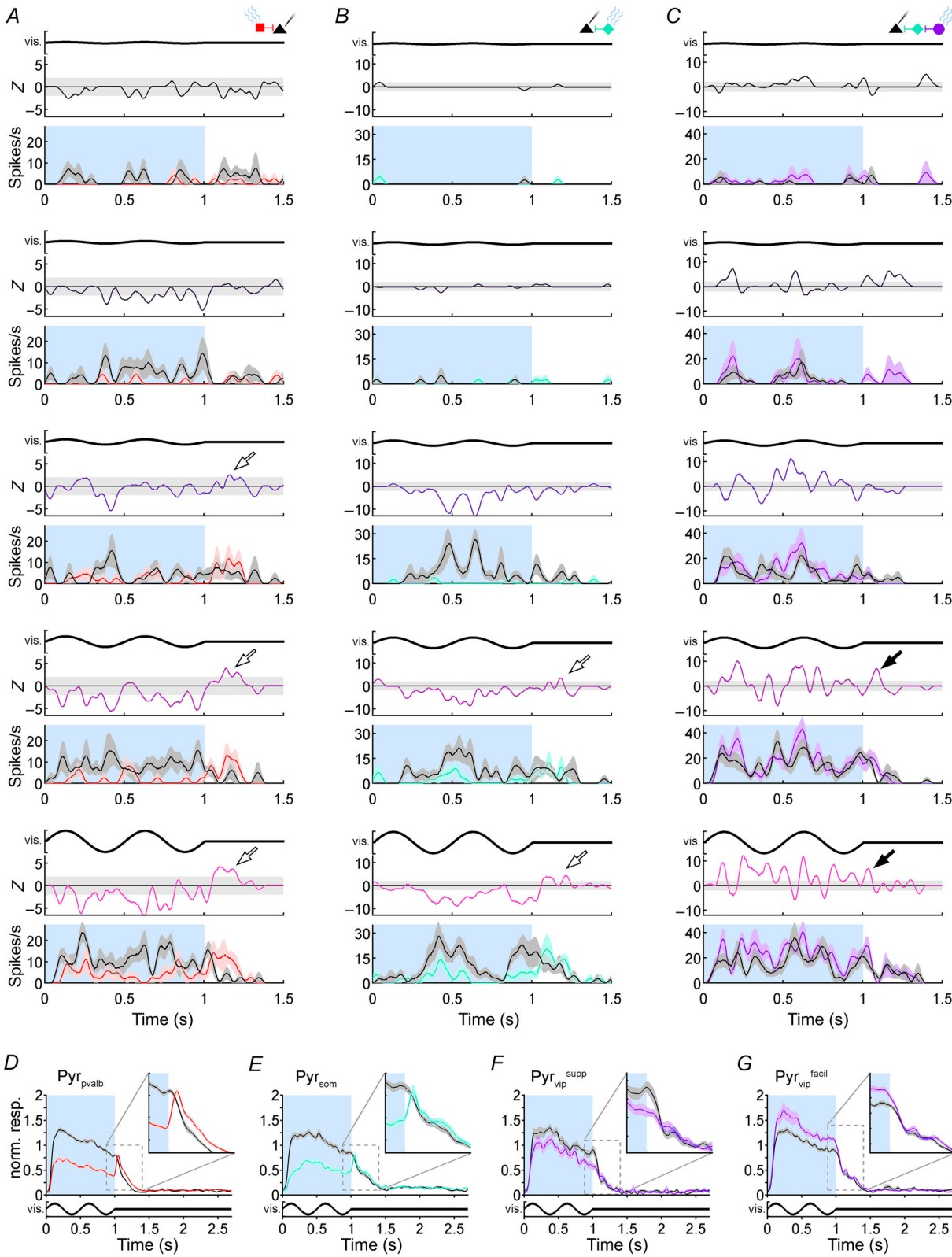

**Figure 2.  Example rebound effects in Pyr cells**

*A*, column of SDFs showing a representative Pyr_pvalb cell's firing during control (black) and Pvalb+ photostimulation (red) conditions during the presentation of gratings with 6, 12, 24, 50 and 100% contrast (top to bottom). For

the drifting grating stimuli (vis.), the bold black line at the top of each graph depicts luminance at one point on the monitor such that the sine wave amplitude indicates contrast. The blue shaded areas (0–1 s) indicate the duration of photostimulation, which temporally overlapped with visual stimulation. The white area in the SDF (1–1.5 s) indicates the post-photostimulation epoch. Z-score functions are shown above each SDF with signature colours for each contrast that are consistent across all figures (6, 12, 24, 50 and 100% contrast as a black to pink gradient). Grey area surrounding the zero-line indicates Z-scores of ±2. *B* and *C*, SDFs and Z-scores showing the responses of representative $Pyr_{som}$ and $Pyr_{vip}^{facil}$ cells during SOM+ and VIP+ photostimulation, respectively. The format is identical to *A*, except the signature colour for each interneuron subtype is taken from Fig. 1*A*. White arrows point to post-inhibitory rebounds in $Pyr_{pvalb}$ and $Pyr_{som}$ cells, and black arrows point to lingering facilitation in the $Pyr_{vip}^{facil}$ cell. *D*, normalized $Pyr_{pvalb}$ population average response to 100% contrast gratings. The bold black line under the graph and blue shaded areas (0–1 s) indicate timing of the visual stimulus and photostimulation, respectively. The dotted box shows the portion of the rebound response that is magnified in the top-right inset panel. *E–G*, normalized population averages for $Pyr_{som}$, $Pyr_{vip}^{supp}$ and $Pyr_{vip}^{facil}$ datasets with the same format as *D*. Shaded areas around SDFs and population averages indicate SEM.

of the grating, and afferent drive remained high in the post-photostimulation epoch due to the grating's continued presence (Fig. 8*A* top-right). In the 'Hi-Lo' condition, the cortex was illuminated between 1001 and 1500 ms, and thus there was high afferent drive when the LED was on due to the presence of the drifting grating, but afferent drive was low in the post-photostimulation epoch due to the grating's disappearance at 1500 ms (Fig. 8*A* top-left). Finally, in the 'Lo-Lo' condition, the cortex was illuminated between 1501 and 2000 ms, and thus there was low afferent drive in both photostimulation and post-photostimulation epochs because the grating had disappeared before the LED turned on (Fig. 8*A* bottom-left). Control and photostimulation trials were randomly interleaved, and we recorded 8–12 repetitions of each condition.

## Data analysis

Isolated single units collected from all three experiments were exported to MATLAB where neuronal responses over time were represented as spike density functions (SDFs) with 1 kHz resolution by convolving a delta function at each spike arrival time with a Gaussian window. The spontaneous activity and maximal firing rates of V1 neurons can vary widely (Niell & Stryker, 2008), so to reliably detect rebounds that were likely to be comparatively small relative to stimulus-evoked activity, we developed a Z-score calculation $[Z = (\varphi - \mu)/\sigma]$ to standardize the magnitude of optogenetic effects in the photostimulation and post-photostimulation epochs. First, we estimated the mean ($\mu$) and standard deviation ($\sigma$) of the difference between photostimulated and control SDFs during a segment of each trial far removed from photostimulation or rebounds effects (the last 1 s of the ISI for Experiment 1; last 100 ms of the ISI for Experiment 2; and last 500 ms of the ISI for Experiment 3). Then, we calculated $\varphi$ as the difference between the photostimulated and control SDFs within the photostimulation and post-photostimulation epochs. These Z-scores were also useful for distinguishing the

ChR2[H134R]-EYFP-expressing interneurons that were rapidly depolarized by light from other neurons that received GABAergic inhibition. We used a threshold Z-score of ±2 to calculate a photomodulation latency when firing in the control and photostimulated conditions began to differ significantly (Shapiro, Gosselin, et al., 2022; Shapiro, Michaud, et al., 2022). Upon cortical illumination, only directly photostimulated interneurons were expected to exhibit low-latency positive Z-scores.

## Statistical analysis

Parametric tests were used for data that appeared approximately normally distributed, and non-parametric analyses were used for clearly skewed data datasets (specific tests noted in the Results). We applied the Benjamini–Hochberg procedure for controlling the false discovery rate (39 total comparisons), and adjusted *P*-values are reported (Benjamini & Hochberg, 1995).

## Results

Our overarching goal was to determine whether optogenetically mediated rebounds were affected by local network activity or activating different members of this network. We performed extracellular recordings in three types of transgenic mice that expressed ChR2[H134R]-EYFP in either Pvalb+, SOM+ or VIP+ interneurons. Prior to searching for rebound effects, we used photomodulation latency and sign to segregate directly photostimulated interneurons from other cells (see Methods). Directly photostimulated interneurons showed rapid depolarization soon after light onset (Fig. 1*B*; Shapiro, Michaud, et al., 2022), whereas post-synaptic neurons could show delayed suppression (Fig. 1*C*) or facilitation (Fig. 1*D*). Across all transgenic mice, every neuron with a photomodulation latency of less than 30ms was facilitated (Z-scores > 2; Fig. 1*E–G*). In Pvalb-Ai32 and Som-Ai32 mice, neurons with longer photomodulation latencies were always suppressed (Z-scores < −2), but neurons recorded in

Vip-Ai32 mice could be suppressed or facilitated at longer latencies. Following Shapiro, Michaud, et al. (2022), we used a photomodulation latency of 30 ms as the border between directly activated interneurons and putative pyramidal neurons (Pyr) that were modulated after some synaptic delay. Across all three experiments we analysed rebounds in 426 $Pyr_{pvalb}$ and 114 Pvalb+ cells recorded in Pvalb-Ai32 mice, 80 $Pyr_{som}$ and 21 SOM+ cells recorded in Som-Ai32 mice, and 99 $Pyr_{vip}$ and 13 VIP+ cells recorded in Vip-Ai32 mice. Putative pyramidal cells in Vip-Ai32 mice were further subdivided into groups that were suppressed ($Pyr_{vip}^{supp}$; $n = 20$) or facilitated ($Pyr_{vip}^{facil}$; $n = 79$). Median photomodulation latencies for each dataset are indicated with arrowheads in Fig. 1*E–G*.

### Experiment 1: contrast effects

In Experiment 1, we searched for rebounds after Pvalb+, SOM+ or VIP+ photostimulation was paired with different levels of afferent drive to V1 that was produced by presenting drifting gratings of different contrasts.

**Pyr cell rebound effects.** We first examined rebound effects in Pyr cells that receive inputs from optogenetically activated interneurons. The example $Pyr_{pvalb}$ cell in Fig. 2*A* showed incrementally larger spike rates to higher visual contrasts in control SDFs (black lines), which were suppressed during Pvalb+ photostimulation (red lines). At the three highest contrast levels, the spike rate in the photostimulated SDF increased above the control SDF shortly after cortical illumination was terminated, signifying rebound spikes (Fig. 2*A*; bottom three SDFs around 1200 ms). When optogenetic modulation was represented as a *Z*-score function (Fig. 2*A*, above each SDF), suppression during the photostimulation epoch produced negative *Z*-scores, and the transiently positive *Z*-scores in the post-photostimulation epoch indicated a rebound effect (Fig. 2*A*, white arrows). For this $Pyr_{pvalb}$ neuron, the rebound effect was largest in the 100% contrast condition. Similar rebound effects were evident in the example $Pyr_{som}$ neuron shown in Fig. 2*B*. Conversely, in the example $Pyr_{vip}^{facil}$ cell shown in Fig. 2*C* wherein VIP+ photostimulation weakly facilitated visually evoked contrast responses, there was some lingering facilitation in the post-photostimulation epoch (black arrows). The average population responses to 100% contrast gratings are shown for $Pyr_{pvalb}$ (Fig. 2*D*; $n = 157$), $Pyr_{som}$ (Fig. 2*E*; $n = 80$), $Pyr_{vip}^{supp}$ (Fig. 2*F*; $n = 20$) and $Pyr_{vip}^{facil}$ (Fig. 2*G*; $n = 79$) datasets, where each neuron's response was normalized by the mean spike rate to 100% gratings. Both $Pyr_{pvalb}$ and $Pyr_{som}$ population averages showed a rapid rebound in firing early in the post-photostimulation epoch, where firing in the photostimulation condition exceeded firing in the control condition, whereas effects in $Pyr_{vip}^{supp}$ and $Pyr_{vip}^{facil}$ datasets appeared smaller.

We summarized rebound effects across each Pyr cell dataset by creating population average *Z*-score functions for $Pyr_{pvalb}$, $Pyr_{som}$, $Pyr_{vip}^{supp}$ and $Pyr_{vip}^{facil}$ neurons at each contrast (Fig. 3*A*, *C*, *E* and *G*). $Pyr_{pvalb}$, $Pyr_{som}$ and $Pyr_{vip}^{supp}$ population data exhibited similar trends to the example cells in Fig. 2*A* and *B*, showing prominent positive rebounds specifically after interneuron photostimulation was paired with 100% contrast stimuli that generated the most afferent drive in V1 (Fig. 3*A*, *C* and *E*). The population average $Pyr_{pvalb}$ rebounds to 100% contrast (Fig. 3*A*) appeared to be about twice the size of rebounds in $Pyr_{som}$ and $Pyr_{vip}^{supp}$ datasets (Fig. 3*C* and *E*). The $Pyr_{pvalb}$ population average also exhibited smaller rebounds at 50% contrast that were not evident in $Pyr_{som}$ or $Pyr_{vip}^{supp}$ datasets (Fig. 3*A*, *C* and *E*). The $Pyr_{vip}^{facil}$ population average produced an entirely different rebound trend compared to suppressed Pyr cells; VIP+ photostimulation produced facilitation (probably via SOM+-mediated disinhibition) that resulted in increased $Pyr_{vip}^{facil}$ firing and positive *Z*-scores during the photostimulation epoch, which waned over time at contrast levels ≤50% (Fig. 3*G*). In the post-photostimulation epoch, there was little evidence of rebounds at the population level following high-contrast stimuli, but for contrasts ≤24% transient positive *Z*-scores after light offset indicated there was an unusual burst of firing following facilitation (Fig. 3*G*).

We quantified the magnitude of rebounds in individual Pyr neurons by calculating the time-averaged *Z*-score from 1050 to 1400 ms at each contrast, which are shown as scatter column plots in Fig. 3*B*, *D*, *F* and *H*. Generally, rebound magnitude increased with stimulus contrast, but there was variability in the size of this effect across datasets. We incorporated all Pyr cell datasets into a mixed-model ANOVA comparing the effect of contrast level (within group) and datasets (between groups) on mean rebound size, which showed significant main effects for Pyr cell rebounds differing between contrast levels ($F[4, 1328] = 2.99$; $P = 0.0413$) and datasets ($F[3, 332] = 7.63$; $P = 2.35 \times 10^{-4}$), with a significant interaction between contrast levels and datasets ($F[12, 1328] = 7.31$; $P = 5.15 \times 10^{-12}$). Tukey honest significant difference (HSD) *post hoc* comparisons indicated mean rebound size across contrast levels for $Pyr_{pvalb}$ cells were significantly larger than for $Pyr_{som}$ cells (MD = 0.25; SE = 0.082; $P_{Tukey} = 1.38 \times 10^{-2}$) and $Pyr_{vip}^{supp}$ cells (MD = 0.53; SE = 0.14; $P_{Tukey} = 1.33 \times 10^{-3}$), but the $Pyr_{som}$ and $Pyr_{vip}^{supp}$ datasets did not significantly differ from each other (MD = 0.28; SE = 0.149; $P_{Tukey} = 0.245$). As expected, the overall pattern of rebounds in $Pyr_{vip}^{facil}$ cells differed greatly from the other datasets, but it is noteworthy that when the ANOVA was re-run excluding the $Pyr_{vip}^{facil}$ dataset both main effects and the interaction

remained significant. Overall, Pyr neurons that were suppressed by interneuron photostimulation produced larger rebounds following high-contrast stimuli that generated the most afferent drive in V1, and Pvalb+

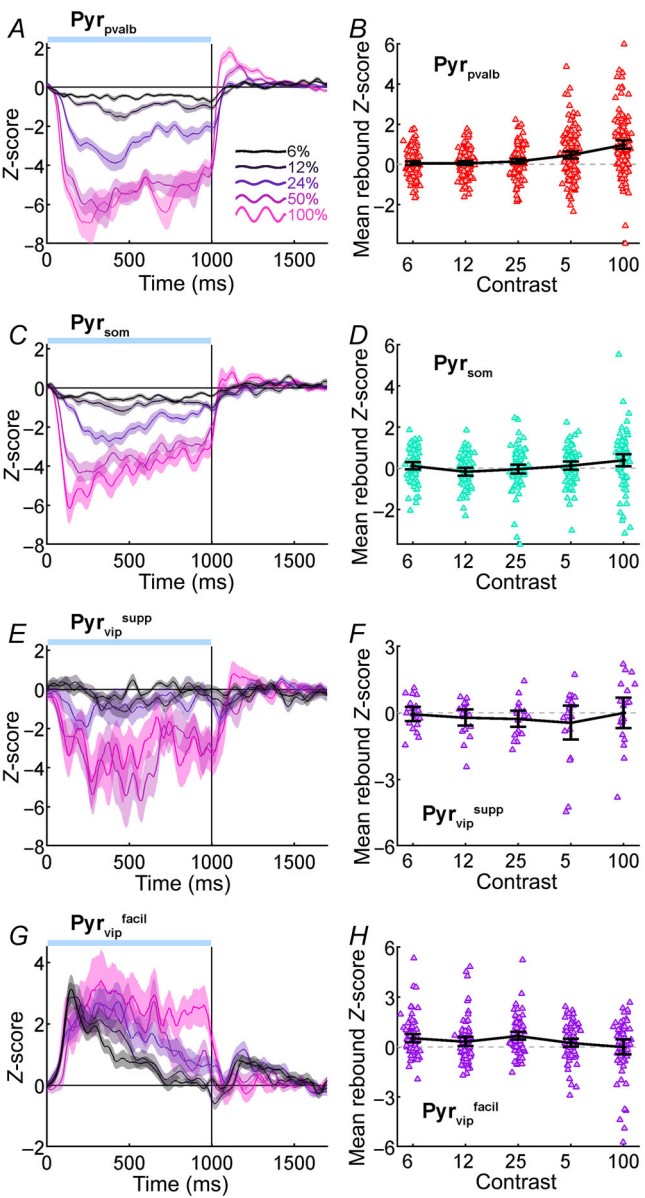

**Figure 3. Population Pyr rebound effects**
*A, C, E* and *G*, population average *Z*-score functions for $Pyr_{pvalb}$, $Pyr_{som}$, $Pyr_{vip}^{supp}$ and $Pyr_{vip}^{facil}$ datasets, respectively. Coloured lines indicate grating contrasts (6, 12, 24, 50 and 100% contrast as a black to pink gradient). Blue bar above each graph indicates the timing of photostimulation. Shaded regions around each population *Z*-score indicate SEM. *B, D, F* and *H*, time-averaged rebound *Z*-scores across each contrast level for $Pyr_{pvalb}$ (red triangles; $n = 157$), $Pyr_{som}$ (teal triangles; $n = 80$), $Pyr_{vip}^{supp}$ (purple triangles; $n = 20$) and $Pyr_{vip}^{facil}$ (purple triangles; $n = 79$) populations, respectively. Dotted horizontal lines indicate *Z*-score of 0. Bold black lines indicate grand average *Z*-score, and error bars indicate 95% confidence intervals.

photostimulation produced the largest rebounds in Pyr cells.

The large dispersion of time-averaged *Z*-scores in Fig. 3*B, D, F* and *H* indicated that rebounds at all contrast levels had highly variable magnitudes. To better visualize variability in rebound size across Pyr cells in a dataset, and for different contrasts within individual Pyr cells, we represented *Z*-score functions as heatmaps then sorted all cells within a dataset by the rebound magnitude at 100% contrast. Figure 4*A* illustrates the $Pyr_{pvalb}$ population data. Each cell number represents an individual $Pyr_{pvalb}$ neuron, plotted in a consistent row across the five heatmaps for the different contrast stimuli. Across-neuron variability within the $Pyr_{pvalb}$ dataset was evident over the rows of all five heatmaps (Fig. 4*A*), but this was most evident for 100% contrast, so we only included 100% contrast heatmaps for $Pyr_{som}$, $Pyr_{vip}^{supp}$ and $Pyr_{vip}^{facil}$ datasets (Fig. 4*D, G* and *J*). Considering first the $Pyr_{pvalb}$, $Pyr_{som}$ and $Pyr_{vip}^{supp}$ datasets where the photostimulation epoch was dominated by suppression, positive rebounds in the post-photostimulation epoch were visible as a yellow streak near the bottom of each 100% contrast heatmap around 1100 ms and were especially prevalent in the $Pyr_{pvalb}$ population (Fig. 4*A, D* and *G*). However, corresponding blue streaks near the top of these same heatmaps depict Pyr cells that continued to be suppressed in the post-photostimulation epoch (Fig. 4*A, D* and *G*), providing clear evidence of across-neuron variability. A similar trend was apparent in the $Pyr_{vip}^{facil}$ dataset, except that the variety of rebounds followed facilitation instead (Fig. 4*J*). For the $Pyr_{pvalb}$ 100% contrast heatmaps, the largest rebounds appeared to be preceded by very strong suppression in the photostimulation epoch, which was supported by a significant moderate negative correlation between suppression and rebound magnitude for $Pyr_{pvalb}$ cells (Fig. 4*B*; $r = -0.41$; $P = 5.40 \times 10^{-7}$). However, this correlation between suppression in the photostimulation epoch and rebound magnitude was not observed for $Pyr_{som}$ (Fig. 4*E*; $r = 0.02$; $P = 0.883$), $Pyr_{vip}^{supp}$ (Fig. 4*H*; $r = 0.34$; $P = 0.248$) or $Pyr_{vip}^{facil}$ cells (Fig. 4*K*; $r = 0.12$; $P = 0.404$). We also checked for potential cortical layer differences by correlating tetrode recoding depth with rebound magnitude, but did not find any significant relationships for $Pyr_{pvalb}$ (Fig. 4*C*; $r = -0.13$; $P = 0.188$), $Pyr_{som}$ (Fig. 4*F*; $r = 3.6 \times 10^{-3}$; $P = 0.975$), $Pyr_{vip}^{supp}$ (Fig. 4*I*; $r = 0.3$; $P = 0.314$) or $Pyr_{vip}^{facil}$ cells (Fig. 4*L*; $r = 0.11$; $P = 0.450$). Finally, we examined the relationship between spike waveform and rebound magnitude because units with narrow spikes are often classified as putative Pvalb+ interneurons (Barthó et al., 2004; DeFelipe et al., 2013, but also see Swadlow 2003). We found that the peak-to-trough intervals of our Pyr cells were significantly bimodal ($P < 0.001$; Hartigan's Dip Statistic bootstrapped for 1000 repetitions), indicating some of our Pyr population could actually be post-synaptic

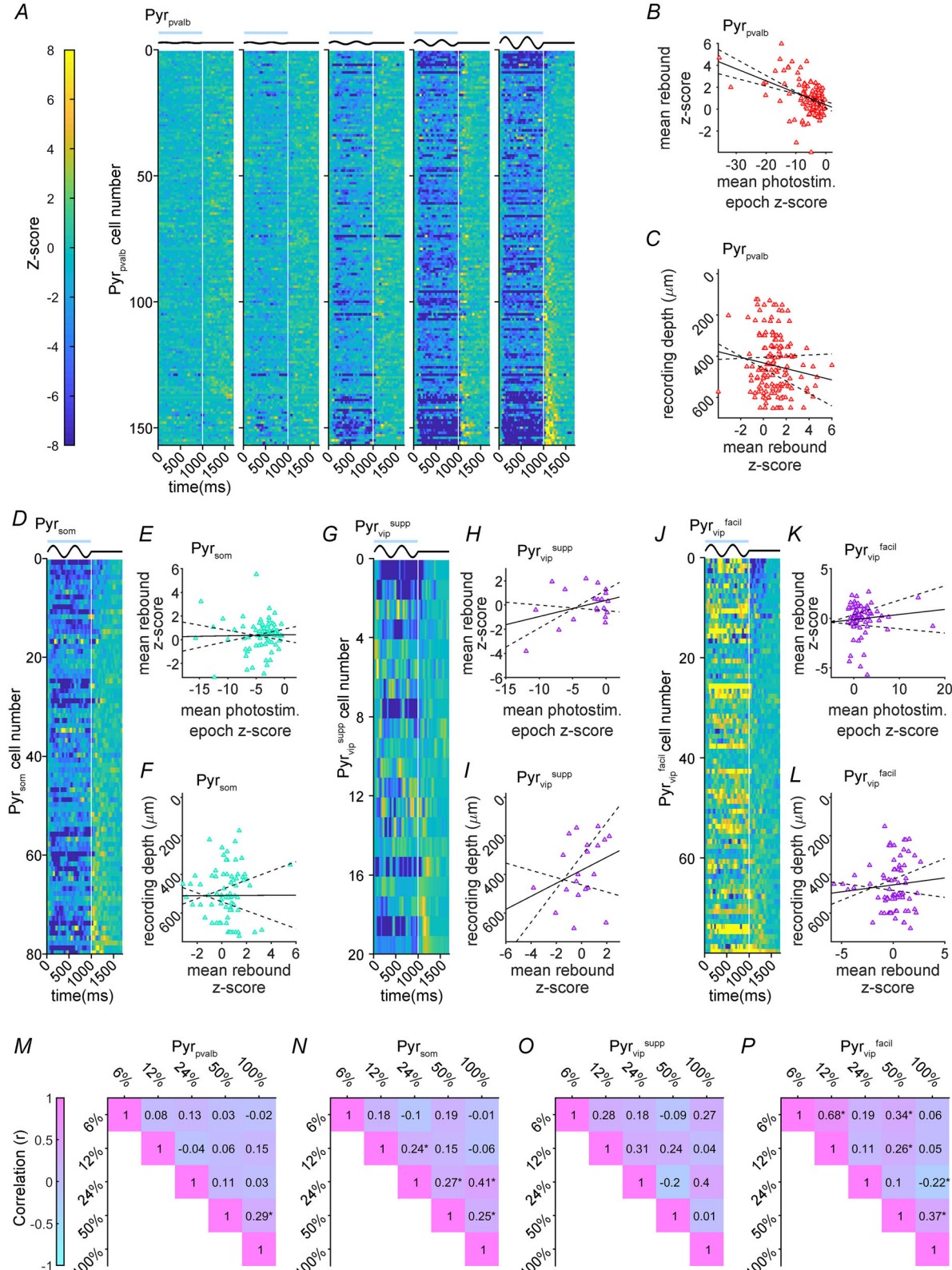

**Figure 4. Variability of rebound size in Pyr cells**

*A*, population heatmaps depicting individual Pyr$_{pvalb}$ *Z*-scores over time (yellow = facilitation; blue = suppression). The five graphs show responses to 6, 12, 24, 50 and 100% contrasts (left to right). Pyr neurons are sorted by

rebound magnitude at 100% contrast, so each row across the five graphs represents the same Pyr$_{pvalb}$ cell. Timing of photostimulation (thick blue lines) and visual stimulation (thick black lines) are shown above each graph. *B*, scatter plot comparing time-averaged *Z*-scores during the photostimulation epoch on the abscissa with time-averaged rebound *Z*-scores on the ordinate for Pyr$_{pvalb}$ cells (red triangles; *n* = 157). *C*, scatter plot comparing time-averaged rebound *Z*-scores on the abscissa with the tetrode recording depth on the ordinate for Pyr$_{pvalb}$ cells (red triangles; *n* = 157). For scatter plots in *B* and *C*, the solid black lines show the linear regression fit to the data and dashed lines show 95% confidence intervals of the fit. Heatmaps only for 100% contrast stimuli and scatter plots are shown for the Pyr$_{som}$ (*D–F*; teal triangles; *n* = 80), Pyr$_{vip}$$^{supp}$ (*G–I*; purple triangles; *n* = 20) and Pyr$_{vip}$$^{facil}$ (*J–L*; purple triangles; *n* = 79) datasets with a format following *A–C*. *M–P*, correlation matrixes comparing the time-averaged rebound magnitude between all contrast levels for Pyr$_{pvalb}$, Pyr$_{som}$, Pyr$_{vip}$$^{supp}$ and Pyr$_{vip}$$^{facil}$ datasets. Numbers in each cell of the matrix indicate the correlation (*r*), and asterisks indicate statistical significance ($P < 0.05$).

Pvalb+ cells receiving inhibition during optogenetic photostimulation. Nevertheless, rebound magnitude was not correlated with peak-to-trough interval for Pyr$_{pvalb}$ ($r = -0.13$; $P = 0.254$), Pyr$_{som}$ ($r = 0.06$; $P = 0.719$), Pyr$_{vip}$$^{supp}$ ($r = 0.17$; $P = 0.637$) or Pyr$_{vip}$$^{facil}$ datasets ($r = 0.02$; $P = 0.883$). Figure 4*A* provides some evidence of within-neuron variability in rebound size between stimulus conditions for the Pyr$_{pvalb}$ dataset because the Pyr$_{pvalb}$ cells that generated the largest rebounds in the 100% contrast heatmap did not always generate rebounds in the 50% or 25% contrast heatmaps. To better assess within-neuron variability, we produced correlation matrixes that determined if rebound size measured by the time-averaged rebound magnitude at one contrast level could predict the rebound size at other levels (Fig. 4*M–P*). Nearly all the correlations for the Pyr$_{pvalb}$, Pyr$_{som}$ and Pyr$_{vip}$$^{supp}$ datasets were weak, with most being not statistically significant. However, the subset of Pyr$_{vip}$$^{facil}$ cells that showed a burst of firing in the post-photostimulation epoch only at lower contrast levels (Fig. 3*G*) produced a significant moderate correlation between rebound sizes at 6% and 12% contrast (Fig. 4*P*). In summary, there was substantial across-neuron and within-neuron variability in rebound magnitude, and only the Pyr$_{pvalb}$ population showed a relationship between suppression in the photostimulation epoch and rebound magnitude in the post-photostimulation epoch.

**ChR2-expressing interneuron post-photostimulation effects.** In our samples of directly photostimulated Pvalb+, SOM+ and VIP+ interneurons we searched for post-excitation modulation akin to the rebounds we found in Pyr cells and examined the influence of afferent drive to V1. The rebounds we observed in our Pyr populations (Figs 2–4) provided an important elaboration on previous work (e.g. Li et al., 2019), so we applied a similar analysis to the photostimulated *ChR2*-expressing interneurons themselves, which have received relatively little attention regarding optogenetically mediated rebound effects. Like Pyr cells, *ChR2*-expressing interneurons in V1 were visually responsive, and showed incrementally larger spike rates to higher contrasts in their control SDFs (Fig. 5*A*; black lines). During photostimulation (red

SDFs), the example Pvalb+ interneuron showed robust increases in firing rates above the control SDF at low contrasts, but more modest increases at higher contrasts (Fig. 5*A*), which has previously been modelled with a saturating additive process (Shapiro, Michaud, et al., 2022). This pattern of modulation generally produced large positive-going *Z*-scores at the beginning of the photostimulation epoch, which tended to decay over time most prominently for higher contrast stimuli (Fig. 5*A* upper panels). Directly after the cortical illumination and visual stimulation were terminated the spike rates in the photostimulation SDF decreased faster than the control SDF, which produced a transiently negative *Z*-score (white arrows in Fig. 5*A*). Increased neural firing that is followed by less firing can be considered a type of negative polarity rebound in spike rate, but to avoid confusion with the wider literature that associates the term 'rebound' with increased spiking, we will refer to this phenomena as post-excitation rapid decay (PERD). For the interneuron shown in Fig. 5*A*, the magnitude of PERD was largest following 100% contrast gratings.

We summarized PERD effects for each interneuron subtype using population average *Z*-score functions (Fig. 5*B*, *D* and *F*). During the photostimulation epoch Pvalb+ and SOM+ population data showed similar trends to the example cell (Fig. 5*B* and *D*), with high-contrast *Z*-score functions waning more than those of lower contrasts. In the post-photostimulation epoch, PERD was most prominent after 100% contrast stimuli that generated the most afferent drive in V1. However, the population average PERD to 100% contrast appeared sharper and deeper for Pvalb+ compared to SOM+ interneurons. VIP+ facilitation during the photostimulation epoch was more similar across contrasts, and following light offset there was no evidence of PERD at the mean population level (Fig. 5*F*). We quantified the magnitude of PERD in individual Pvalb+, SOM+ or VIP+ interneurons by calculating the time-averaged *Z*-score from 1010 to 1100 ms at each contrast, which were then plotted as scatter column graphs in Fig. 5*C*, *E* and *G*. Pvalb+ and SOM+ scatter column plots showed PERD magnitude becoming increasingly negative as contrast increased (Fig. 5*C* and *E*), but this trend was not evident in our small

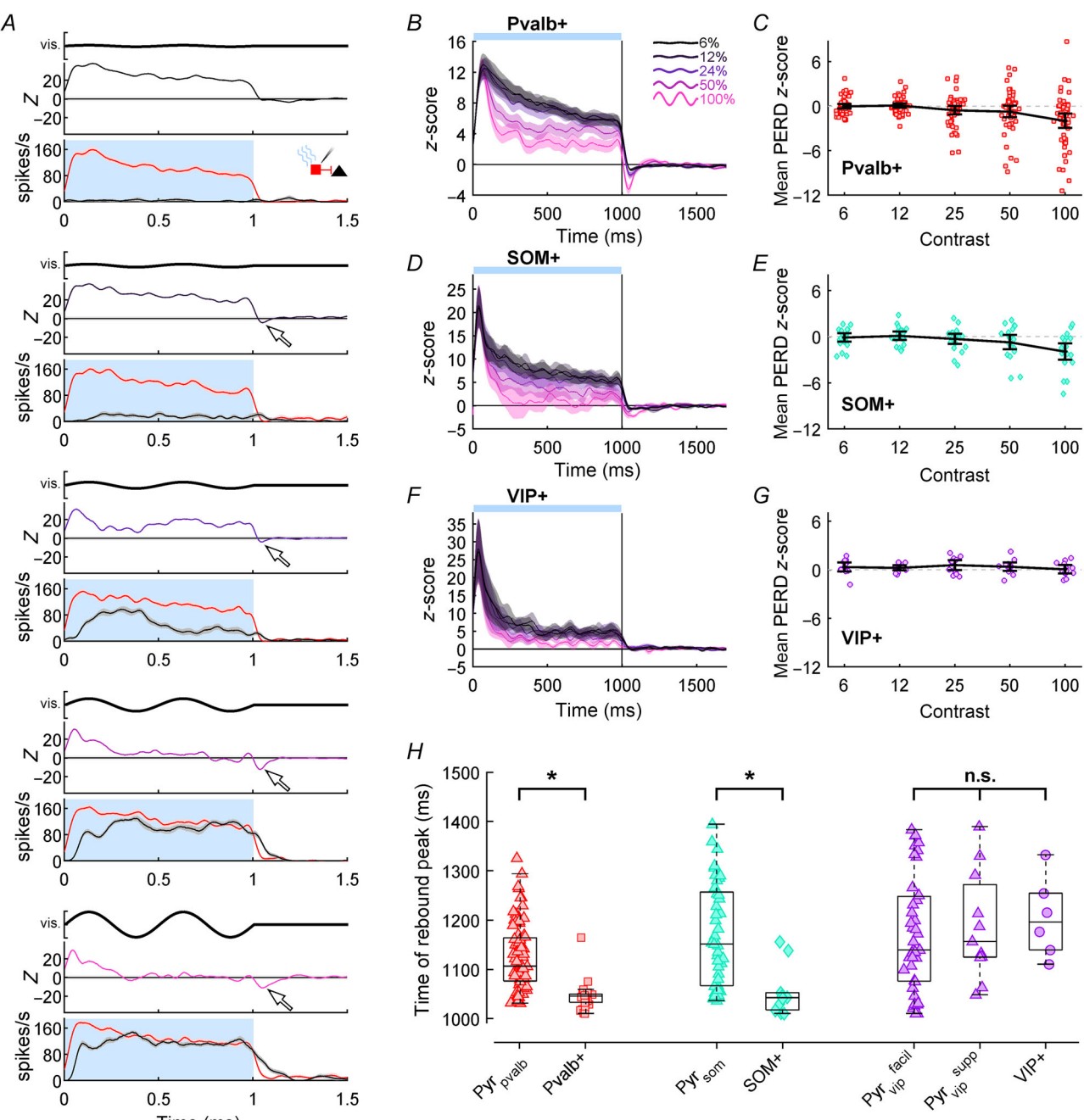

**Figure 5. PERD effects in photostimulated interneurons**

*A*, column of graphs with SDFs and *Z*-scores showing a representative Pvalb+ interneuron's firing during control (black) and photostimulation (red) conditions during the presentation of gratings with 6, 12, 24, 50 and 100% contrast (top to bottom). Format follows Fig. 2*A*. *B*, *D* and *F*, population average *Z*-score functions for Pvalb+, SOM+ and VIP+ datasets following the format of Fig. 3*A*. Shading around each *Z*-score trace indicates SEM. *C*, *E* and *G*, time-averaged PERD *Z*-scores across each contrast level for Pvalb+ (red squares; *n* = 54), SOM+ (teal diamonds; *n* = 21) and VIP+ populations (purple circles; *n* = 13), respectively. Dotted horizontal lines indicate *Z*-score of 0. Bold black lines indicate grand average *Z*-score, and error bars indicate 95% confidence intervals. *H*, scatter-column plots comparing the latency of rebound peaks for Pyr cells and interneurons in Pvalb-Ai32 (Pyr$_{pvalb}$: *n* = 79; Pvalb+: *n* = 27), Som-Ai32 (Pyr$_{som}$: *n* = 41; SOM+: *n* = 10) and Vip-Ai32 (Pyr$_{vip}^{facil}$: *n* = 39; Pyr$_{vip}^{supp}$: *n* = 11; SOM+: *n* = 6) mice. Boxplots indicate population medians and quartiles. Brackets denote a comparison with a significant (*) or non-significant (n.s.) difference (see Results).

sample of VIP+ cells (Fig. 5*G*). PERD magnitude was highly variable across all interneuron subtypes, showing positive and negative *Z*-scores at each contrast (Fig. 5*C*, *E* and *G*). We incorporated all interneuron datasets into a mixed-model ANOVA comparing the effect of contrast level (within group) and interneuron subtype (between groups) on mean PERD size, which showed a significant main effect for PERD amplitude differing between contrasts ($F[4, 340] = 7.13$; $P = 6.97 \times 10^{-5}$), but no main effect of interneuron subtype ($F[2, 85] = 2.73$; $P = 0.139$), and no significant interaction between contrast levels and interneuron subtype ($F[8, 340] = 1.05$; $P = 0.534$). Thus, interneurons generally showed larger PERD after their photostimulation was paired with higher contrast stimuli, but some of our qualitative observations of differences between interneuron subtypes were not captured by this time-averaged quantitative analysis.

The most obvious disparity between rebound effects in our Pyr cells (Figs 2 and 3) and PERD in interneurons (Fig. 5) was the difference in directionality. However, considering Pyr cells and interneurons are reciprocally connected (Isaacson & Scanziani, 2011), and photo-modulation latencies are shorter in *ChR2*-expressing interneurons than Pyr cells (Fig. 1), we speculated interneuron PERD may occur before Pyr cell rebounds. For population *Z*-scores at 100% contrast, Pvalb+ and SOM+ interneurons very rapidly reached negative values after light offset (Fig. 5*B* and *D*), whereas suppressed Pyr cells appeared to take slightly longer to reach positive values (Fig. 3*A*, *C* and *E*). We quantified this timing difference by calculating the latency to the rebound and PERD peaks, but this analysis required clearly demarcated rebounds/PERD, so we only included the top half of each dataset that produced the largest rebound or PERD amplitudes to 100% contrast gratings (Fig. 5*H*). Mann–Whitney *U* tests indicated that the PERD produced by Pvalb+ and SOM+ interneurons reached their respective peaks significantly earlier than the rebounds produced by Pyr$_{pvalb}$ ($T(104) = 5.98$; $P = 7.08 \times 10^{-9}$) and Pyr$_{som}$ cells ($T(49) = 3.39$; $P = 1.73 \times 10^{-3}$). Considering the small and infrequent rebounds observed in Vip-Ai32 mice, it was not surprising that a Welch's ANOVA found no evidence of a difference in the timing of VIP+, Pyr$_{vip}$$^{supp}$ or Pyr$_{vip}$$^{facil}$ rebounds ($F[2, 13.5] = 0.394$; $P = 0.739$). Thus, there was convincing evidence that PERD preceded rebounds only for interneuron subtypes that send direct projections to Pyr cells.

Our interneuron datasets showed highly variable PERD magnitudes at each contrast much like the Pyr datasets (compare Fig. 3*B*, *D*, *F* and *H* with Fig. 5*C*, *E* and *G*), so we used population heatmaps to illustrate across- and within-interneuron variability in Fig. 6. Figure 6*A* represents each Pvalb+ interneuron in a consistent row across the five heatmaps for the different contrast stimuli. Across-interneuron variability was easiest to see in the

heatmaps for 100% contrast stimuli shown for the Pvalb+ (Fig. 6*A*, right), SOM+ (Fig. 6*D*) and VIP+ datasets (Fig. 6*G*). Pvalb+ and SOM+ interneuron PERDs in the post-photostimulation epoch were visible as a blue streak near the top of each heatmap around 1050 ms. However, several Pvalb+ interneurons at the bottom of the 100% contrast heatmap showed the opposite effect and continued to be facilitated in the post-photostimulation epoch. VIP+ cells, even in their 100% contrast heatmap, rarely generated clear PERD (Fig. 6*G*). Many of the Pvalb+ and SOM+ interneurons that showed strong saturation, or even paradoxical suppression, in the photo-stimulation epoch at high contrasts also showed the strongest PERD (cells in the top few rows in 50–100% contrast heatmaps). In fact, there was a significant moderate positive correlation between modulation during the photostimulation epoch and PERD magnitude for Pvalb+ cells (Fig. 6*B*; $r = 0.59$, $P = 1.37 \times 10^{-5}$) and SOM+ cells (Fig. 6*E*; $r = 0.54$, $P = 3.12 \times 10^{-2}$). However, for VIP+ cells PERD and lingering facilitation were quite small and about equally common (Fig. 6*G*), and there was not a significant correlation between modulation during the photostimulation epoch and PERD magnitude (Fig. 6*H*; $r = -0.34$, $P = 0.390$). Shapiro, Michaud, et al. (2022) examined the contrast response functions of optogenetically activated inter-neurons and found that Pvalb+ and SOM+ interneurons recorded in superficial cortical layers showed photo-stimulation effects with more saturation at high contrasts, which prompted us to examine cortical layer differences for the post-photostimulation epoch too. Pvalb+ inter-neurons showed a significant weak positive correlation between tetrode recoding depth and PERD magnitude (Fig. 6*C*; $r = 0.33$, $P = 0.0375$), which combined with the results from Fig. 6*B* indicated that Pvalb+ cells recorded from superficial layers tended to show more saturation in the photostimulation epoch and larger PERD effects. However, correlations between recoding depth and PERD were not significant for SOM+ (Fig. 6*F*; $r = 0.13$, $P = 0.719$) or VIP+ cells (Fig. 6*I*; $r = 0.15$, $P = 0.723$). The Pvalb+ interneuron heatmaps in Fig. 6*A* also illustrated within-neuron variability, with some of the cells producing strong PERD only at 100% while others consistently showed PERD over 24–100% contrasts. We quantified within-neuron variability for Pvalb+, SOM+ and VIP+ interneurons by creating correlation matrixes that determined if time-averaged PERD magnitude at one contrast level could predict PERD size at other levels (Fig. 6*J–L*). The Pvalb+ dataset exhibited several significant, moderate correlations (Fig. 6*J*), but most correlations for SOM+ and VIP+ datasets were not significant (Fig. 6*K* and *L*). Overall, there was considerable across- and within-interneuron variability in PERD magnitude, but it was distinct from the trends observed in Pyr datasets. Collectively, the data from Experiment

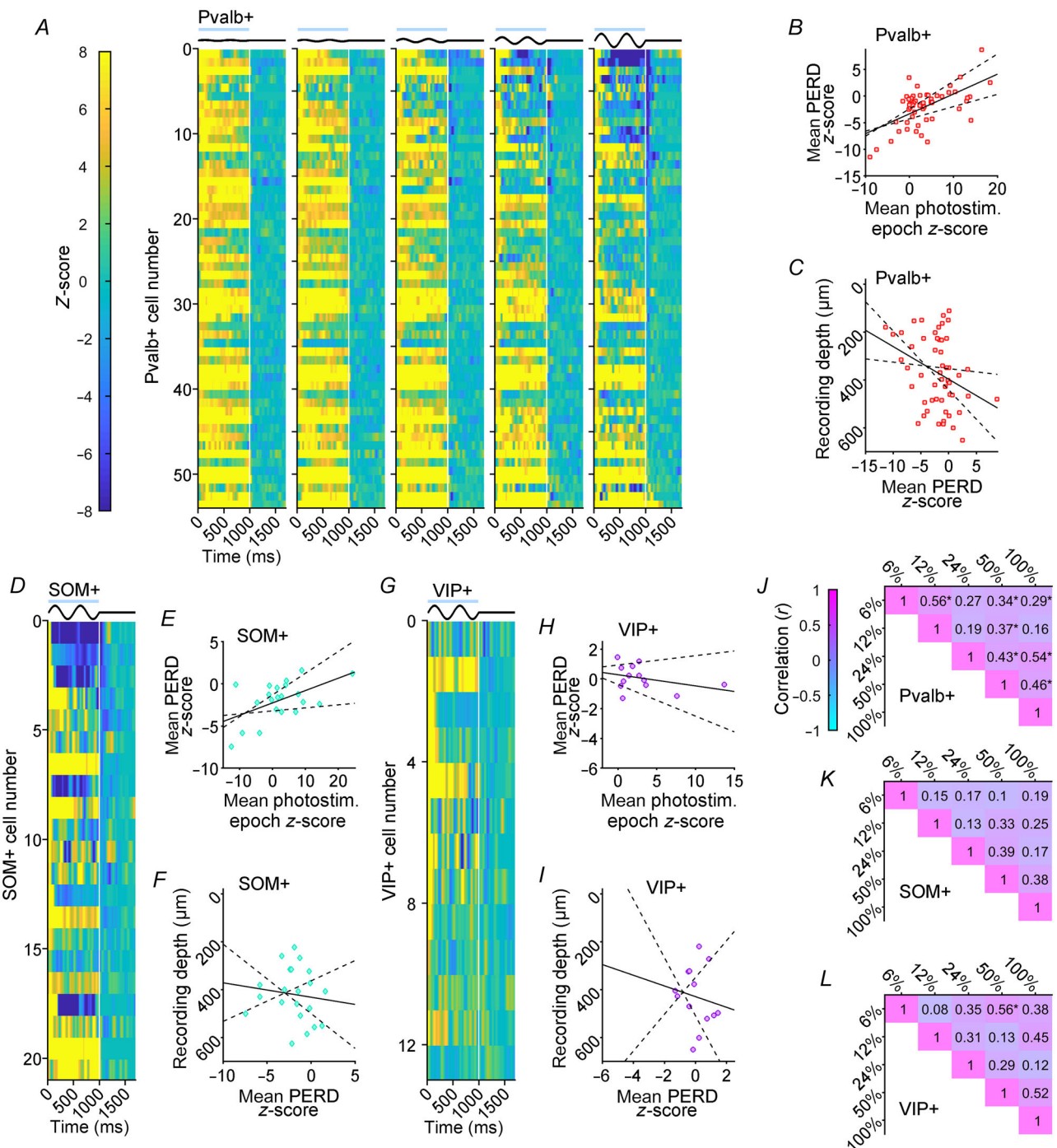

**Figure 6. Variability of PERD in photostimulated interneurons**

*A*, population heatmaps depicting *Z*-scores from individual Pvalb+ interneurons over time in an identical format to Fig. 4*A*. *B*, scatter plot comparing time-averaged *Z*-scores during the photostimulation epoch on the abscissa with time-averaged PERD *Z*-scores on the ordinate for Pvalb+ interneurons (red squares; *n* = 54). *C*, scatter plot comparing time-averaged PERD *Z*-scores on the abscissa with the tetrode recording depth on the ordinate for Pvalb+ interneurons. For scatter plots in *B* and *C*, the solid black lines show the linear regression fit to the data and dashed lines show 95% confidence intervals of the fit. Heatmaps only for 100% contrast stimuli and scatter plots are shown for the SOM+ (*D–F*; teal diamonds; *n* = 21) and VIP+ (*G–I*; purple circles; *n* = 13) interneurons with a format following *A–C*. *J–L*, correlation matrixes comparing the time-averaged PERD magnitude between all contrast levels for Pvalb+, SOM+ and VIP+ datasets. Numbers in each cell of the matrix indicate the correlation (*r*), and asterisks indicate statistical significance (*P* < 0.05).

1 showed that rebounds in Pyr cells and PERD in inter-neurons were usually largest following visual stimuli that strongly activate V1 during photostimulation, and that despite variability in all datasets the rebounds following Pvalb+ photostimulation tended to be the largest.

## Experiment 2: spatial effects

In Experiment 2 we aimed to corroborate our findings that high afferent drive to V1 increases rebound magnitude by testing if this effect could be reproduced with different visual stimuli. Considering that $Pyr_{pvalb}$ cell rebounds in Experiment 1 were strongest and most abundant (Fig. 3), we decided to pair Pvalb+ photostimulation with vertically oriented black or white bars that were randomly flashed within or outside V1 RFs (Fig. 7*A* and *B*). Data were analysed from 143 $Pyr_{pvalb}$ cells.

Figure 7*A* shows SDFs of an example complex cell's response to white bars flashed in nine different spatial locations during control (black lines) or Pvalb+ photo-stimulated conditions (red lines). In the control condition, bars flashed within the RF elicited robust onset responses and sometimes offset responses to the appearance and disappearance of the bar, respectively. Spiking responses were much weaker when bars were flashed closer to the edges of the RF, and bars flashed outside the RF only elicited spontaneous firing. During photostimulated trials, onset responses were attenuated when the cortex was illuminated, but spike rates in the post-photostimulation epoch were potentiated, producing rebound effects at bar positions inside the RF (Fig. 7*A*). We next converted SDFs into *Z*-score functions, such that suppression during the photostimulation epoch appeared as negative *Z*-scores, and rebounds during the post-photostimulation epoch appeared as transient positive *Z*-scores (Fig. 7*C* and *D*).

To examine whether afferent drive during the photo-stimulation epoch affected later rebound size, for each cell we compared the *Z*-score function produced by the bars flashed inside the RF that generated the strongest visual response with a *Z*-score function produced by bars flashed outside the RF that generated the weakest visual response (Fig. 7*D*). Across the population, we compared *Z*-score functions inside *versus* outside the RF in response to black bars for 139 $Pyr_{pvalb}$ cells and to white bars for 143 $Pyr_{pvalb}$ cells. Figure 7*E* and *F* plot the population average *Z*-score functions for white and black bars, respectively. The population showed robust rebounds following light offset for both white and black bars flashed inside the RF (green lines), but not for bars flashed outside the RF (black lines). We quantified the difference in rebounds produced by bars flashed inside *versus* outside the RF by comparing the time-averaged *Z*-scores between 700 and 900 ms, which was the period rebounds were largest. Most $Pyr_{pvalb}$ cells had larger time-averaged *Z*-scores when bars

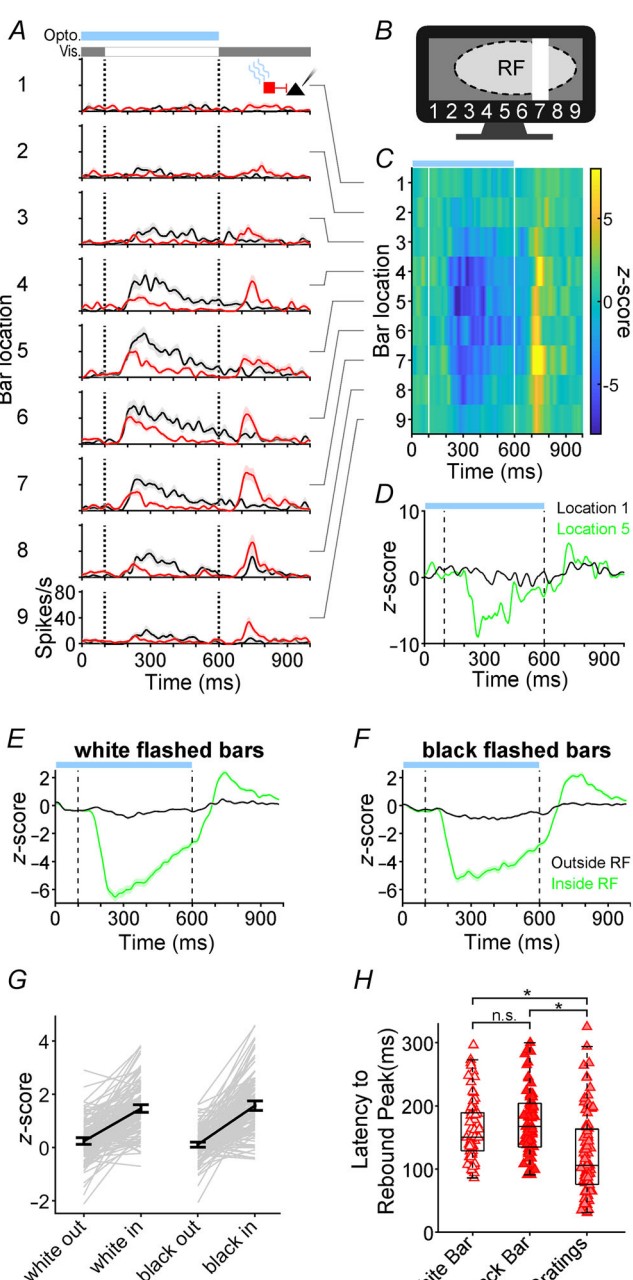

**Figure 7. Rebound effects to flashed bars**
*A*, SDFs from a representative Pyr cell showing response to white bars flashed in spatial locations 1–9 without (black) or with Pvalb+ photostimulation (red). The timing of photostimulation (Opto.; blue bar) and the visual stimulus (Vis.) are shown above the SDFs. Shaded area around each SDF indicates SEM. *B*, cartoon of the visual stimulus for Experiment 2 showing an example trial where a white bar is flashed at position 7 inside the neuron's RF (dashed oval). *C*, *Z*-score heatmaps of each bar location for the neuron in *A*. *D*, comparison of *Z*-score traces from bar locations eliciting the strongest (lime) and weakest visual response (black) from the example Pyr cell in *A*. *E*, population average *Z*-score functions for white flashed bars comparing bar locations eliciting strong responses inside the RF (lime) and weak responses outside the RF (black). Photostimulation timing is shown by the blue bar at the top of the

graph. Vertical dashed lines indicate when the flashed bar stimulus appeared and disappeared. Shaded regions surrounding the *Z*-score lines indicate SEM. *F*, population response to flashed black bars following the same format as *E*. *G*, time-averaged *Z*-scores for white (*n* = 143) and black (*n* = 139) bars flashed inside *versus* outside RFs. Individual cells are shown by light grey lines, whereas black lines show population averages, and error bars indicate 95% confidence intervals. *H*, scatter-column plot comparing Pyr rebound peak latency for white bars (*n* = 72), black bars (*n* = 70) and drifting gratings (*n* = 79). Boxplots indicate population medians and quartiles. Brackets denote comparisons with a significant (*) or non-significant (n.s.) difference (see Results).

were flashed inside the RF (Fig. 7*G*; grey lines), and this was also reflected by the population means (Fig. 7*G*; black lines with error bars). A repeated-measures ANOVA [bar location (inside RF *vs.* outside RF) × stimulus polarity (white bars *vs.* black bars)] showed a main effect of bar location indicating bars flashed inside RFs produced significantly larger rebounds than bars flashed outside RFs ($F[1, 130] = 338$; $P = 1.00 \times 10^{-36}$), with no evidence of a difference between black and white bars ($F[1, 130] = 0.2$; $P = 0.729$), or any interaction between bar location and stimulus polarity ($F[1, 130] = 3.5$; $P = 0.129$). These results from Experiment 2 corroborated our findings from Experiment 1, suggesting that increasing afferent drive during the photostimulation epoch potentiates later rebound size during the post-photostimulation epoch.

The rebounds in Experiment 2 often coincided with transient visual offset responses to bar disappearance (e.g. Fig. 7*A*), which seemed to occur later than the rebounds created by the facilitation that followed drifting gratings in Experiment 1 (e.g. Fig. 2*A*). This timing difference was also evident in the population average *Z*-score functions from each experiment (compare Figs 3*A* with 7*E* and *F*). Therefore, we conducted an additional analysis examining latency to rebound peak for Pyr$_{pvalb}$ cells following presentation of flashed white or black bars and 100% contrast drifting gratings. Again, we used the top half of each dataset that displayed the largest rebound amplitudes to ensure rebound peaks could be detected accurately. A Kruskal–Wallis one-way ANOVA showed the timing of rebound peaks did differ significantly [$\chi^2(2) = 31.9$; $P = 6.45 \times 10^{-7}$; Fig. 7*H*]. *Post hoc* Dwass–Steel–Critchlow–Fligner pairwise comparisons indicated white and black bar peak rebound latencies did not differ significantly from each other ($W = 2.16$; $P = 0.279$), but drifting grating peak rebound latencies were significantly shorter than peak rebound latencies for both black ($W = -7.19$; $P = 1.11 \times 10^{-6}$) and white bars ($W = -6.19$; $P = 3.54 \times 10^{-5}$). This disparity in rebound timing suggests that the nature of afferent drive during and after photostimulation can produce subtle differences in rebounds.

## Experiment 3: timing effects

A common theme in the design of both Experiments 1 and 2 was that the optogenetic photostimulation and visually driven activity both terminated at the same time. In Experiment 3 we sought to determine whether this co-occurrence was required to produce rebounds, and whether manipulating the level of afferent drive before, during or after the photostimulation epoch influenced rebound size. We used a factorial design where visually driven neural activity could be high (Hi) or low (Lo) during the photostimulation and post-photostimulation epochs (Fig. 8*A*; see Methods). For this experiment we again solely collected data in Pvalb-Ai32 mice, and report findings from a population of 134 Pyr$_{pvalb}$ and 34 Pvalb+ cells.

**Pyr cell rebound effects.** Experiments 1 and 2 showed converging evidence that rebounds were largest in Pyr$_{pvalb}$ cells following the pairing of photostimulation with strong afferent drive to V1 (Figs 2–4 and 7). Therefore, for Experiment 3 we predicted that conditions where Pvalb+ photostimulation co-occurred with high afferent drive (Fig. 8*A*; HiHi and HiLo) should generally produce larger rebounds than conditions where Pvalb+ photostimulation co-occurred with low afferent drive (Fig. 8*A*; LoHi and LoLo). However, we were most interested in comparisons between conditions with high *versus* low afferent drive during the post-photostimulation epoch (e.g. HiHi *vs.* HiLo, and LoHi *vs.* LoLo), which had not been examined before. Figure 8*A* shows data from an example Pyr cell with spiking responses in control (black lines) and photostimulation (red lines) conditions plotted as SDFs for all four conditions. In response to the HiLo condition, which most resembles Experiment 1, this Pyr cell showed a small and transient elevation in rebound spiking when the visual stimulus and photostimulation simultaneously concluded (Fig. 8*A*, top left). Rebounds produced from both HiHi and LoHi conditions with high afferent drive in the post-photostimulation epoch appeared to produce prolonged and complex trains of rebound spiking (Fig. 8*A*, top and bottom right). Finally, this cell did not produce noticeable rebounds for the LoLo condition, where there was low afferent drive in both photostimulation and post-photostimulation epochs (Fig. 8*A*, bottom left).

We examined all four conditions across our population with average *Z*-score functions (Fig. 8*B–E*) and population heatmaps (Fig. 8*G*), but now each condition was temporally aligned with the photostimulation epoch at the start of each graph. As expected, the population data showed more robust suppression in the photostimulation epoch when it coincided with visual stimulation (Fig. 8*B*, *C* and *G*; HiLo and HiHi), compared to when Pvalb+ activation occurred before (Fig. 8*E* and *G*; LoHi) or after

the visual stimulus (Fig. 8*D* and *G*; LoLo). During the post-photostimulation epoch, the population average for the HiLo condition showed transient rebounds reminiscent of those described in Experiment 1 but with lower magnitude (Fig. 8*B*). In the population heatmaps, the frequent but mild rebounds appeared as a pale-yellow streak that extended from the bottom of the HiLo column (Fig. 8*G*). Despite the similarity in photostimulation offset relative to the visual stimulus, rebound magnitudes in the Experiment 3 HiLo condition were significantly smaller than Pyr$_{pvalb}$ rebounds in Experiment 1 ($P = 9.23 \times 10^{-9}$; Student's *t* test), which aligns well with previous findings

that photostimulation lasting 0.5 s produced rebounds about half as large as photostimulation lasting 1 s (Li et al., 2019). In comparison, the HiHi condition showed prolonged and complex rebounds after light offset, with many individual cells showing stronger excitation or inhibition lasting over 500 ms (Fig. 8*G*). The population average showed small oscillatory rebound activity right after light offset that appeared to grow and reach a peak around the time of visual stimulus offset that occurred ∼500 ms later (Fig. 8*C*). The population average for the LoHi condition oscillated around zero after light offset (Fig. 8*E*), which was potentially due to the low

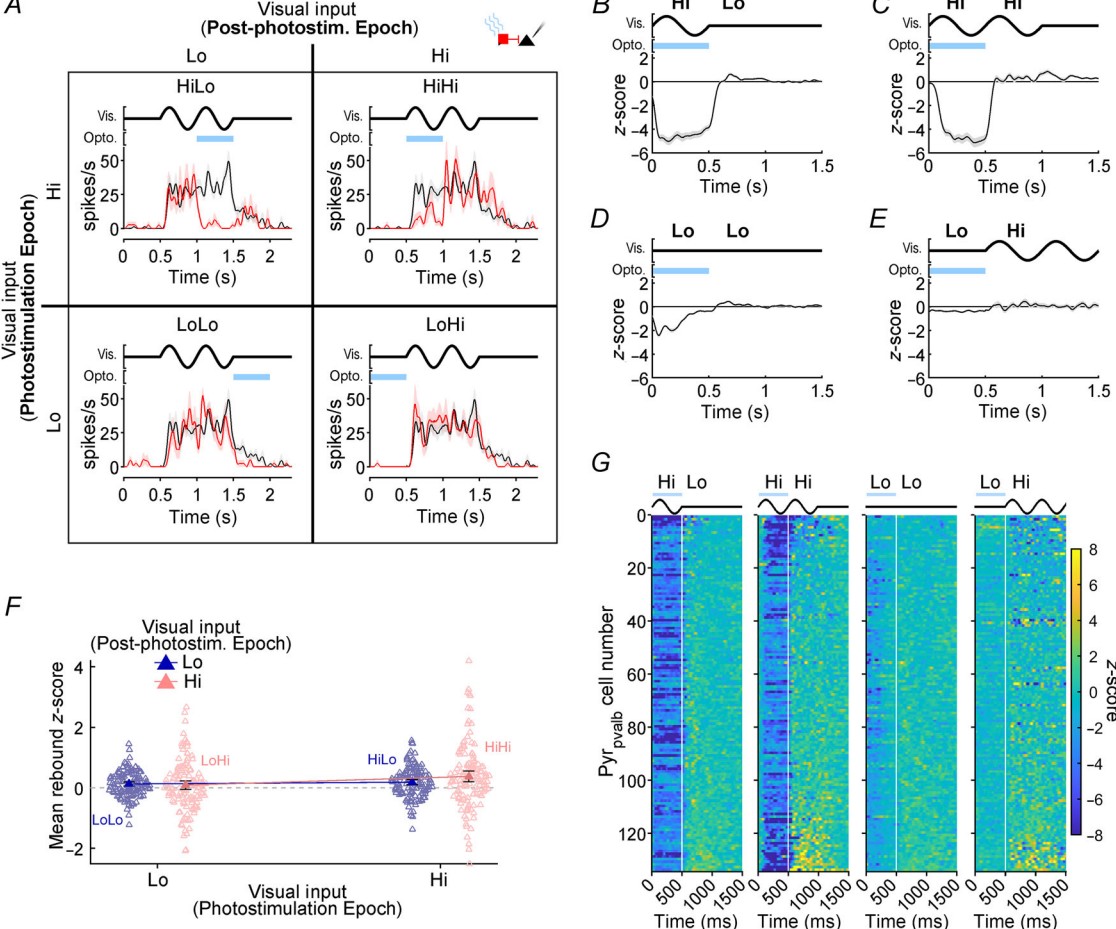

**Figure 8. Pyr$_{pvalb}$ rebounds measured with a factorial design**
*A*, factorial design to test the effect of high (Hi) *versus* low (Lo) visual drive during the photostimulation and post-photostimulation epochs with conditions arranged in a Punnett square style (see Methods). SDFs for each condition show the responses of the same Pyr$_{pvalb}$ neuron during control (black) and photostimulation (red) trials. Shaded regions surrounding SDFs indicate SEM. *B–E*, population average *Z*-score functions for each condition. Shading surrounding black traces indicates SEM. *F*, solid triangles indicate estimated marginal means of time-averaged rebound *Z*-scores when post-photostimulation epoch visual drive was Hi (pink) or Lo (violet). Empty triangles show individual datapoints ($n = 134$). Error bars indicate 95% confidence intervals. *G*, population heatmaps depicting individual ($n = 134$) Pyr$_{pvalb}$ *Z*-score functions over time (yellow = facilitation; blue = suppression). The four graphs show responses in HiLo, HiHi, LoLo and LoHi conditions (left to right), with Pyr$_{pvalb}$ neurons sorted by rebound magnitude separately for each condition. For *A–E* and *G*, the timing of photostimulation (Opto.) and the visual stimulus (Vis.) are indicated at the top of each graph by the blue bar and bold black line, respectively.

level of afferent drive in the photostimulation epoch, and because the smaller prolonged rebounds and inhibition of individual neurons cancelled each other out (Fig. 3*G*). The LoLo population average showed small and transient rebounds (Fig. 8*D*), perhaps because afferent drive during the photostimulation epoch was slightly higher than in the LoHi condition due to the gradual decay in firing that occurs after drifting gratings disappear. To test whether Pyr cell rebounds were significantly impacted by the different levels of afferent drive during both epochs, we calculated time-averaged *Z*-scores between 100 and 700 ms of the post-photostimulation epoch (Fig. 8*F*). A factorial ANOVA (afferent drive in the photostimulation epoch × afferent drive in the post-photostimulation epoch) showed a main effect of afferent drive in the photostimulation epoch indicating rebounds were significantly greater following a period where high visually evoked activity was strongly suppressed by Pvalb+ activation ($F[1, 133] = 9.36$; $P = 8.71 \times 10^{-3}$). There was no evidence for a main effect of afferent drive in the post-photostimulation epoch affecting rebound size ($F[1, 133] = 1.56$; $P = 0.333$) or an interaction between afferent drive in the photostimulation and post-photostimulation epochs ($F[1, 133] = 4.45$; $P = 0.0799$). However, the significant main effect of activity during the photostimulation epoch appears to mainly be driven by the prolonged rebounds of the HiHi condition (e.g. compare transient and prolonged rebounds in Fig. 8*B* and *C*, respectively). Pyr$_{pvalb}$ cell data in Experiment 3 substantiated data in Experiments 1 and 2, demonstrating again that strongly driving V1 during the photostimulation epoch with highly effective visual stimuli increased rebound size after light offset. However, Experiment 3 also showed that the co-termination of optogenetic photostimulation and visually driven activity was not required to produce rebounds, and that afferent drive in the post-photostimulation epoch could alter rebound timing and amplitude in complex ways.

**Pvalb+ PERD effects.** Based on our observations in Experiment 1 that PERD size for *ChR2*-expressing Pvalb+ interneurons was also affected by afferent drive (Figs 5 and 6), we predicted conditions in Experiment 3 where Pvalb+ photostimulation co-occurred with high afferent drive (Fig. 9*A* and *B*; HiLo and HiHi) should generally produce larger PERD than conditions where Pvalb+ photostimulation co-occurred with low afferent drive (Fig. 9*C* and *D*; LoLo and LoHi). However, as with our Pyr data in Experiment 3 (Fig. 8), we were most interested in comparisons between conditions with Hi *versus* Lo afferent drive during the post-photostimulation epoch because the influence of afferent drive during the PERD was unknown. Figure 9*A*–*D* show data from an example Pvalb+ interneuron with spiking responses shown as control (black lines) and photostimulation SDFs (red lines) for all four conditions. This Pvalb+ interneuron showed substantial activation from photostimulation in all four conditions. However, when cortical illumination was combined with visual stimulation (Fig. 9*A* and *B*; HiLo and HiHi) the difference in firing between control and photostimulation responses was smaller than when cortical illumination occurred before (Fig. 9*D*; LoHi) or after visual stimulation (Fig. 9*C*; LoLo), which resembled the saturating additive facilitation seen in the interneurons from Experiment 1 (Fig. 5*A*). In response to the HiLo condition, which most resembled Experiment 1, this Pvalb+ interneuron showed a small and transient decrease in firing when the visual stimulus and photostimulation simultaneously concluded, producing a PERD (Fig. 9*A*). For the HiHi condition, post-photostimulation epoch firing quickly decreased below the control levels, also producing a PERD (Fig. 9*B*). However, this Pvalb+ cell produced a much smaller PERD in the LoHi condition (Fig. 9*D*), potentially because in the photostimulation condition the drop in firing was masked by a strong onset transient produced by the grating's appearance, or because the interneuron's spike rate in the control condition was still low due to its visual response latency. The PERD was also minuscule in the LoLo condition (Fig. 9*C*), probably because the firing rate of the Pvalb+ interneuron was already very low at this time.

We examined all four conditions across our interneuron population with average *Z*-score functions (Fig. 9*E*–*H*) and population heatmaps (Fig. 9*J*), which were again temporally aligned to the photostimulation epoch. As predicted from Experiment 1 and the example cell, the population data showed waning facilitation during the photostimulation epoch when cortical illumination coincided with visual stimulation (Fig. 9*E* and *F*; HiLo and HiHi), compared to when Pvalb+ activation occurred before (Fig. 9*H*; LoHi) or after the visual stimulus (Fig. 9*G*; LoLo). The population average for the LoHi condition showed a PERD that was predominantly negative, but also prolonged, low amplitude and complex (Fig. 9*H*). The LoHi heatmap also shows many cells had prolonged negative *Z*-scores in the post-photostimulation epoch, although some cells exhibited a mix of positive and negative *Z*-scores over time (Fig. 9*J*). In comparison, both the HiLo and HiHi conditions showed sharp PERD after light offset (Fig. 9*E* and *F*), although the HiHi PERD was about twice as large probably because more interneurons in this condition showed strong PERD (Fig. 9*J*). Finally, the LoLo condition showed virtually no PERD (Fig. 9*G* and *J*). To quantify whether Pvalb+ PERD was significantly impacted by the level of afferent drive during both epochs we calculated time-averaged *Z*-scores between 20 and 400ms of the post-poststimulation epoch (Fig. 9*I*). A factorial ANOVA (afferent drive in the photostimulation epoch × afferent

drive in the post-photostimulation epoch) showed main effects for both afferent drive during the photostimulation epoch ($F[1, 33] = 8.8$; $P = P = 1.54 \times 10^{-2}$) and post-photostimulation epoch ($F[1, 33] = 9.9$; $P = 1.06 \times 10^{-2}$), but no evidence of an interaction ($F[1, 33] = 0.25$; $P = 0.723$). This indicated high afferent drive in both the photostimulation and post-photostimulation epochs increased the amplitude of PERD. However, when afferent drive during the post-photostimulation epoch was high (Fig. 9*I*, pink symbols) the estimated marginal means showed larger error bars, indicating this condition also increased variability in PERD size across the population. Overall, Pvalb+ cell data in Experiment 3 corroborated data from Experiment 1 by reinforcing the finding that strongly driving V1 during the photostimulation epoch increased PERD amplitude after light offset, but added a new finding that the co-termination of optogenetic photostimulation and visually driven activity

was not required to produce PERD, and in fact even larger PERD magnitudes could be revealed when afferent drive in the post-photostimulation epoch was high.

## Discussion

In this study we investigated how optogenetically mediated rebounds in mouse V1 were affected by the activity level of the local network, and whether activating different interneuron subtypes embedded within this network produced distinct rebound effects. Our results indicate that coupling interneuron photostimulation with high visual drive in V1 was most conducive for producing rebound effects in Pyr cells and PERD in *ChR2*-expressing interneurons. Also, Pvalb+ photostimulation produced more frequent and larger rebounds than SOM+ or VIP+ photostimulation. However, rebound effects were quite variable across all datasets. Below, we compare these

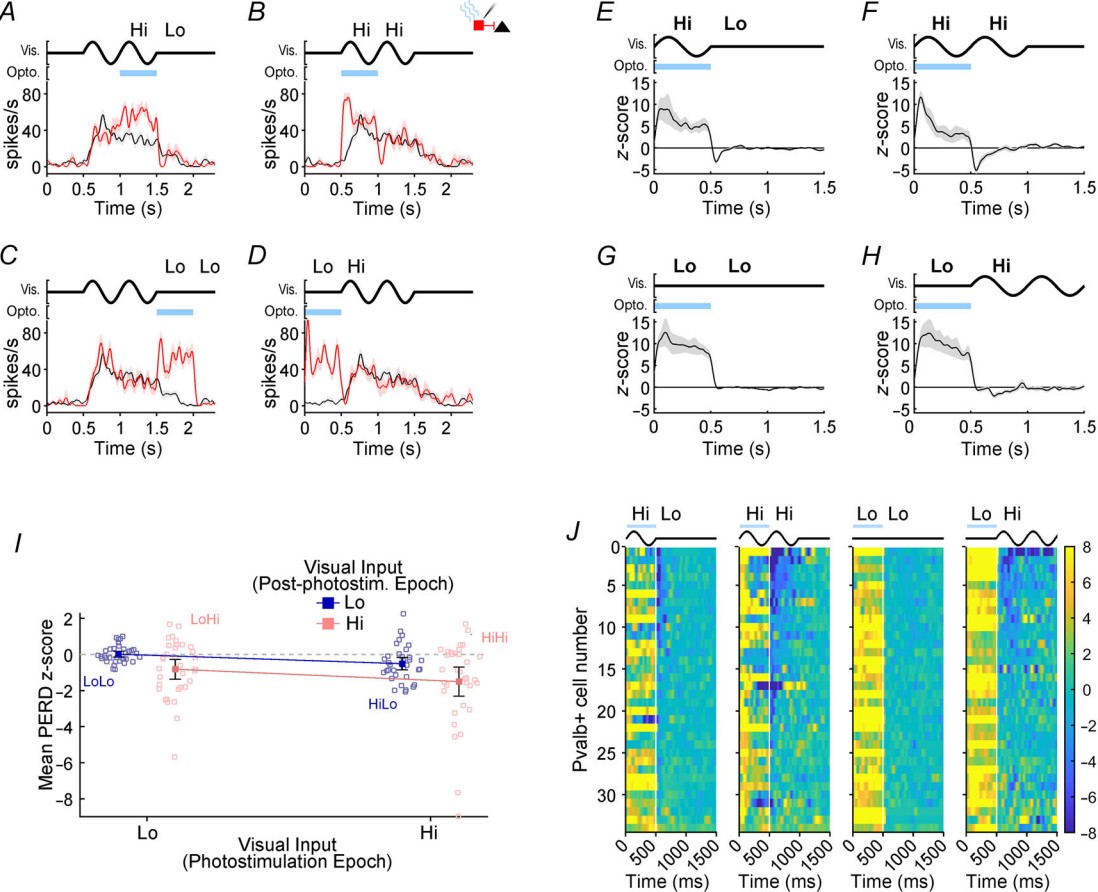

**Figure 9. Pvalb+ PERD measured with a factorial design**
*A–D*, an example Pvalb+ cell's firing during control (black) and photostimulation (red) trials from all four conditions in the factorial design following the format of Fig. 8*A* (see Methods). *E–H*, population average *Z*-score functions for each condition following the same format as Fig. 8*B*. *I*, solid squares indicate estimated marginal means of time-averaged PERD *Z*-scores when post-photostimulation epoch visual drive was Hi (pink) or Lo (violet). Empty squares show individual datapoints ($n = 34$). Error bars indicate 95% confidence intervals. *J*, population heatmaps depicting individual ($n = 34$) Pvalb+ *Z*-score functions following the same format as Fig. 8*G*.

results with past rebound literature and discuss cellular and network mechanisms that may underly rebound effects in mouse V1.

### Prevalence of rebounds

A potential role for rebound effects in subcortical oscillatory circuits and central pattern generators has been appreciated for some time (Marder & Calabrese, 1996; McCormick & Bal, 1997), but with the increasingly common application of optogenetic techniques it seems that rebounds can be produced (and may play a physiologically relevant role) in many additional brain areas. However, it is challenging to estimate how common these rebounds are in the literature because most studies that reported optogenetically mediated rebounds merely described their presence (Ayzenshtat et al., 2016; Chang et al., 2018; Jin & Glickfeld, 2020; Madisen et al., 2012; Sessolo et al., 2015; Tønnesen et al., 2009; Tsunematsu et al., 2011) or methods for limiting them (Chuong et al., 2014; Mahn et al., 2016), whereas optogenetic studies more rigorously examining rebound effects are scarcer (Lado et al., 2022; Li et al., 2019; Stark et al., 2013). The literature also contains apparent instances of PERD following photostimulation of pyramidal cells (Cone et al. 2020, their fig. 5b) or GABAergic interneurons (Luis-Islas et al. 2022, their fig. 1d), which have received little attention.

Fortunately, it appears that the prevalence of rebound activity does not depend heavily on the analytical method used. Our results show that rebounds and PERD were readily detectable using raw spike rates (as in the example cells in Figs 2*A–C*, 5*A*, 7*A*, 8*A* and 9*A*), whether responses were normalized by baseline firing or maximum firing (Fig. 2*D–G*), or using *Z*-scores (Figs 3–9). Importantly, even though we favoured the *Z*-score calculation, it was not required to observe our results. We believe normalizing data using the *Z*-score calculation is the most sensible approach because it takes variability in spiking into account when measuring the difference between photostimulation and control conditions. For example, a relatively small increase in firing in the post-photostimulation epoch (compared to the large visually evoked responses) may not be important if the baseline firing is highly variable, but is more likely to be important if the baseline firing has low variability.

A more serious challenge for assessing the prevalence of optogenetically mediated rebounds is their variability. The first issue is that it seems probable that several mechanisms underlie rebound effects (Calabrese & Feldman, 1999), so not every cell type or neural circuit will produce optogenetically mediated rebounds across all experimental conditions. We found extensive rebound variability in our datasets measured *in vivo* under anaesthesia (Figs 4 and 6), but variability has also been reported *in vitro* with cortical slice preparations (Madisen et al. 2012). The second issue is that the magnitude and prevalence of optogenetically mediated rebounds varies widely based on several experimental parameters. We found that under the most conducive conditions of high afferent drive some form of rebound was detectable in approximately 25% (Vip-Ai32 mice) to 75% (Pvalb-Ai32 mice) of recorded neurons. Conversely, in conditions with low afferent drive, rebounds overall were much smaller and so rare they could easily be overlooked. Photostimulation parameters have also been shown to affect rebounds. For example, Li et al (2019) systematically varied the power (1.5–7 mW) and duration (500–4000 ms) of their photostimulating laser and found that rebounds were largest at maximum power and duration. This finding corroborates earlier work indicating increasing the hyperpolarizing pulse amplitude or duration could increase rebound strength (Aizenman & Linden, 1999; Johnson & Getting, 1991). Interestingly, the power of the LED photostimulation we used was comparatively weaker than past optogenetic studies that reported rebounds *in vitro* (Chang et al., 2018; Lado et al., 2022; Madisen et al., 2012; Mahn et al., 2016; Sessolo et al., 2015; Tønnesen et al., 2009), or those that used *in vivo* methods in spontaneously active cortical neurons (Li et al., 2019). Future work seeking to combine sensory and optogenetic stimulation could examine whether much stronger photostimulation might increase rebound prevalence, mask the effect of afferent drive or reduce across-neuron variability of rebound magnitude in cortical neurons.

### Cellular rebound mechanisms

In some neurons, hyperpolarization below −65 mV can activate $I_h$ currents, which subsequently activates low-threshold $Ca^{2+}$ currents and drives post-inhibitory rebound spiking (Biel et al., 2009; Perez-Reyes, 2003). However, this $I_h$ mechanism is not consistent with the results from our Pyr cells where rebounds were larger following robust visually evoked activity (Figs 3 and 7). We did not perform intracellular recordings, but the spike threshold non-linearity is sufficiently well characterized in mouse V1 (e.g. Tan et al., 2011) to surmise that strong visual stimulation should depolarize Pyr cells well above their spike threshold, which would have led to de-activation and inactivation of $I_h$ and low-threshold $Ca^{2+}$ currents, respectively (Wahl-Schott & Biel, 2009). The role of $I_h$ in pacemaking is now thought to be minimal except for a few neuronal types (Bean, 2025), so perhaps our results could be better explained by persistent $Na^+$ current now more associated with burst firing and pacemaking (Brumberg et al., 2000; Taddese & Bean, 2002). Persistent $Na^+$ current refers to the tiny proportion TTX-sensitive $Na^+$ channels that remain open

at depolarized voltages in equilibrium with a much larger fraction of inactivated $Na^+$ channels. Persistent $Na^+$ current is modestly reduced by high-frequency spiking (Do & Bean, 2003), which aligns with our finding that activity affects rebound magnitude. During weak visual drive, where there is little difference between control and photostimulated Pyr responses (i.e. divisive inhibition; Wilson et al. 2012), there would be similar amounts of persistent $Na^+$ current and post-inhibition firing. During strong visual drive, the photostimulation condition would have lower firing but greater persistent $Na^+$ current compared to the control condition to produce more depolarization and generate rebound spikes when the inhibition was terminated. Studying rebounds in V1 has several methodological advantages, so it would be beneficial for future work to clarify the cellular rebound mechanisms prevalent in this brain region.

The PERD observed in photostimulated interneurons could be partially explained by spike-frequency adaptation and slow after-hyperpolarization mediated by $Ca^{2+}$-activated and/or $Na^+$-activated $K^+$ currents that often follows prolonged neural activation (Constanti & Sim, 1987; Madison & Nicoll, 1984; Pennefather et al., 1985; Pineda et al., 1998; Sah, 1996; Sanchez-Vives et al., 2000; Schwindt et al., 1988, 1989, 1992; Storm, 1993). Optogenetic activation of ChR2[H134R] cation channels in our visually driven interneurons would probably increase the influx of $Na^+$ and $Ca^{2+}$ to further activate outward $K^+$ currents and magnify spike-frequency adaptation and slow after-hyperpolarization to produce a faster decay in spike rate compared to the control condition (e.g. Figs 5*A* and 9*A–D*; Nagel et al., 2005). Furthermore, a floor effect produced by the spike threshold non-linearity would make this decay appear larger with high afferent drive as we observed. Intrinsic differences in the propensity for generating spike-frequency adaptation or slow after-hyperpolarization could underly the PERD differences we found between interneuron subtypes. Advanced optical methods to perform targeted intracellular recordings of genetically tagged neurons could provide further evidence for these intrinsic rebound mechanisms while also elucidating differences across neuron subtypes (e.g. Annecchino et al., 2017).

### Network rebound mechanisms

The Pyr cell rebounds observed in Experiments 1–3 seem better explained by persistent $Na^+$ currents than $I_h$ as an intrinsic cellular mechanism, but they are also broadly consistent with the network mechanism of release from inhibition (Singer, 1996). Pvalb+ and SOM+ interneurons send direct inhibition to Pyr cells (Fig. 1*A*; Karnani et al., 2016; Pfeffer et al., 2013), so at the beginning of the post-photostimulation epoch the rapid decline in firing by Pvalb+ and SOM+ interneurons may temporarily decrease inhibitory post-synaptic potentials in the Pyr cells and subsequently allow rebound depolarization and spikes. Conversely, for VIP+ cells that have indirect polysynaptic connections to Pyr cells (Karnani et al., 2016; Pfeffer et al., 2013), release from inhibition would predict complex or weak rebound effects in Pyr cells, which is also consistent with our results (Figs 3 and 4). Finally, each interneuron subtype we studied showed across-neuron variability of PERD amplitudes (Fig. 6), which could produce varying amounts of release from inhibition and potentially contribute to the across-neuron variability of Pyr post-inhibitory rebounds.

Feedback inhibition is a distinct network mechanism that could contribute to PERD. For inhibition-stabilized network models of cortical circuits, unstable local excitation is regulated by strong feedback inhibition, whereby pyramidal cells first activated by thalamic input thereafter excite interneurons that then inhibit these same pyramidal cells (Isaacson & Scanziani, 2011; Fig. 1*A*). Mass optogenetic activation of interneurons can suppress Pyr firing so much that it chokes off Pyr excitation to these same interneurons leading them to fire less, which is called paradoxical inhibition (Li et al., 2019; Miller 2016; Ozeki et al., 2009; Tsodyks et al., 1997). PERD could arise if the firing rates of interneurons remained artificially elevated by optogenetic photostimulation, then when photostimulation was terminated, and with no Pyr excitation to keep them depolarized, the firing rate of these interneurons would drop precipitously. Again, targeted intracellular recordings of the various interneuron subtypes (Annecchino et al., 2017), and potentially *in vivo* recordings of monosynaptically connected neuron pairs (Jouhanneau & Poulet, 2019), would be useful to disentangle network from cellular mechanisms leading to rebounds and PERD.

Finally, it is important to consider whether the connectivity patterns of Pvalb+, SOM+ and VIP+ interneurons could contribute to the differences in rebound prevalence and strength. For example, although both Pvalb+ and SOM+ interneurons directly inhibit Pyr cells, they differ in several ways including their distribution and input across cortical lamina (Bortone et al., 2014; Xu et al., 2010), their local network connectivity (Fig. 1*A*; Karnani et al., 2016; Pfeffer et al., 2013), the regions of Pyr cells they target (Fig. 1*A*; Tremblay et al., 2016), their afferent input from thalamocortical cells (Ji et al., 2016) and how they summate excitatory inputs (Kapfer et al., 2007; Silberberg, 2008; Silberberg & Markram, 2007). Pvalb+ interneurons are specialized to produce fast and transient inhibition (Cardin et al., 2009; Tremblay et al., 2016), and individual Pvalb+ interneurons evoke larger unitary inhibitory post-synaptic currents in Pyr cells than do SOM+ cells (Safari et al., 2017). Thus, this

tight temporal control of firing may have produced the sharper and deeper PERD in the Pvalb+ interneurons themselves, and a precisely timed and powerful release from inhibition effect to elicit larger rebounds in our Pyr$_{pvalb}$ dataset. Furthermore, the anatomical and physiological subdivisions within SOM+ and VIP+ categories are more heterogeneous than for Pvalb+ (Jiang et al., 2015; Tremblay et al., 2016), which may have led rebounds in Pvalb-Ai32 mice to be more uniform and therefore appear larger at the population level. VIP+ interneurons' polysynaptic and disinhibitory connections to Pyr cells make their network contributions to rebounds more difficult to interpret. For example, using Archaerhodopsin to photo-inhibit VIP+ cells (i.e. opposite to the manipulation we performed) can elicit post-inhibitory rebounds in Pyr cells virtually identical to those we found in our Pyr$_{vip}^{supp}$ dataset (Ayzenshtat et al., 2016; Lado et al., 2022).

### Anaesthesia and rebounds

When comparing past work with our findings in isoflurane-anaesthetized mice to speculate on potential mechanisms underlying optogenetically mediated rebounds there are two effects of anaesthesia to consider. The first issue relates to the intrinsic cellular processes that produce rebounds, where we recognized that several types of anaesthetics target HCN channels and reduce $I_h$ (e.g. isoflurane: Chen et al., 2009; sevoflurane: Schwerin et al., 2021; enflurane: Tokimasa et al., 1990; halothane: Sirois et al., 1998; xenon: Mattusch et al., 2015; pentobarbital: Wan et al., 2003; ketamine: Zhou et al., 2013; propofol: Chen et al. 2005). Critically, the concentration of inhalational isoflurane required to avoid suppressing visually evoked activity for *in vivo* physiology (0.2–0.6% typically supplemented with chlorprothixene; Frantz et al. 2020; Nsiangani et al. 2022; Smith & Häusser, 2010) is substantially lower than the nominal concentrations of anaesthetic perfused onto brain slices for *in vitro* studies (1–2%; Chen et al., 2009; Tennigkeit et al., 1997; Timic Stamenic et al., 2019). Furthermore, the *in vitro* evidence that isoflurane has a dose-dependent effect on low-threshold $Ca^{2+}$ spikes and rebounds suggests $I_h$ currents should be only mildly affected by the 0.5% isoflurane anaesthesia we used *in vivo* (Ries & Puil, 1999). Even so, if we consider the hypothetical unlikely scenario where $I_h$ is critical for rebounds in V1 but is fully blocked by 0.5% isoflurane, there is still an explanation required for why we, and several others (Chuong et al., 2014; Lee et al., 2012), observed optogenetically mediated rebounds under isoflurane anaesthesia. In this scenario, the alternative explanations we discussed in the *Cellular rebound mechanisms* and *Network rebound mechanisms* sections above could be interpreted as additional processes that were uncovered under $I_h$ blockade. The second issue is that anaesthesia induced unconsciousness

changes V1 responses, which could alter the magnitude or repeatability of optogenetically mediated rebounds. Overall, V1 classical RF properties are similar between awake and anaesthetized mice (Durand et al., 2016; Pisauro et al. 2013). However, neural recordings in awake mice often have a high level of trial-to-trial variability sometimes referred to as 'noise', which has now been attributed to cortex-wide signals related to spontaneous movements, choice behaviour on operant tasks and arousal/attention (Musall et al. 2019; Salkoff et al. 2020; Stringer et al. 2019). Furthermore, during this mixing of sensory, motor and attentional information within V1, the amplitude of non-visual signals can be up to 4-fold larger than responses to visual stimuli (Abdolrahmani et al., 2021). Optogenetically mediated rebounds in cortex have been characterized in awake but quiescent mice (Li et al., 2019), but examining the relationship between neural activity and rebound magnitude as we have done here could be more challenging in awake mice due to the unpredictable timing and amplitude of synaptic inputs to the recorded neurons relative to optogenetic modulation. Thus, the low spontaneous firing rate observed under anaesthesia may have made it easier for us to detect rebounds with mild photostimulation and more critically to demonstrate the change in rebound magnitude with sensory drive. Ultimately, the trade-off regarding anaesthesia is between control and realism: investigating brain function under anaesthesia emphasizes simplification by reducing the known and as yet unexplainable interactions that could affect neural activity in order to isolate particular phenomena, whereas awake behaving recordings prioritize realism by giving up some control to embrace the noisy complexity of the conscious brain.

### Conclusions

Overall, we found that when optogenetically mediated rebounds were studied within an active cortical network, rebound magnitude could depend on both the level of network activity and which neural subtypes were optogenetically manipulated. Clearly, the variety and complexity of these rebound effects merit further investigation. With the increasing prevalence of ever more powerful optogenetic tools applied to the study of cortical circuits, more attention should be given to rebound effects to help understand their mechanisms and impacts. Optogenetically mediated rebounds can also provide insights into brain function if the rebounds and PERD we observed are considered exaggerated versions of normal processes induced by an artificially large or prolonged bout of inhibition or excitation, respectively. *In vitro* brain slice studies have used this same logic for examining post-inhibitory rebound spiking to characterize intrinsic cellular mechanisms that could regulate depolarization and hyperpolarization within a neuron to generate

pacemaking or rhythmic burst firing (Marder & Calabrese, 1996; McCormick & Bal, 1997; Satterlie, 1985; Wahl-Schott & Biel, 2009). On a more macro scale, the balance of inhibition and excitation has also been a central theme in studies of cortical network function (Isaacson & Scanziani, 2011), and it seems plausible that the cellular and network mechanisms that produce rebounds following optogenetic perturbation would under normal circumstances act to regulate neural and/or network excitability within a narrow operating range.

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

## Additional information

### Data availability statement

The datasets and code that support the findings of this study are available from N.A.C. upon reasonable request.

### Competing interests

The authors declare that they have no competing interests or conflicts of interest to disclose.

### Author contributions

J.S.: Investigation, Formal analysis, Writing – Original Draft. N.M.: Formal analysis, Writing – Original Draft. N.C.: Conceptualization, Software, Resources, Formal analysis, Writing – Original Draft, Supervision, Funding acquisition.

### Funding

This work was supported by the Natural Sciences and Engineering Research Council of Canada (RGPIN/06761-2015, RGPIN/03810-2023) and Canada Foundation for Innovation (24436).

### Keywords

electrophysiology, interneurons, neural circuits, post-inhibitory rebound effect, transgenic mouse

## Supporting information

Additional supporting information can be found online in the Supporting Information section at the end of the HTML view of the article. Supporting information files available:

**Peer Review History**

