## [Peer Review History · The Journal of Physiology]

Characterizing Optogenetic-mediated Rebound Effects in Anesthetized Mouse Primary Visual Cortex

Jared T Shapiro, Nicole M Michaud, and Nathan A Crowder

DOI: 10.1113/JP287265

Corresponding author(s): Nathan Crowder (nathan.crowder@dal.ca)

The following individual(s) involved in review of this submission have agreed to reveal their identity: Michael Okun (Referee #2)

Review Timeline:

Submission Date:	13-Jul-2024
Editorial Decision:	27-Aug-2024
Revision Received:	30-Apr-2025
Editorial Decision:	30-May-2025
Revision Received:	12-Jun-2025
Accepted:	24-Jun-2025

Senior Editor: Nathan Schoppa

Reviewing Editor: Conny Kopp-Scheinflug

Transaction Report:

Dear Dr Crowder,

Re: JP-RP-2024-287265 "Characterizing Optogenetic-mediated Rebound Effects in Mouse Primary Visual Cortex" by Jared T Shapiro, Nicole M Michaud, and Nathan A Crowder

Thank you for submitting your manuscript to The Journal of Physiology. It has been assessed by a Reviewing Editor and by 2 expert referees and we are pleased to tell you that it is potentially acceptable for publication following satisfactory major revision.

REVISION CHECKLIST:

Please upload two versions of your manuscript text: one with all relevant changes highlighted and one clean version with no

changes tracked. The manuscript file should include all tables and figure legends, but each figure/graph should be uploaded as separate, high-resolution files.

We look forward to receiving your revised submission.

Yours sincerely,

Nathan Schoppa
Senior Editor
The Journal of Physiology

REQUIRED ITEMS

- Include a Key Points list in the article itself, before the Abstract.
- Author photo and profile. First or joint first authors are asked to provide a short biography (no more than 100 words for one author or 150 words in total for joint first authors) and a portrait photograph. These should be uploaded and clearly labelled together in a Word document with the revised version of the manuscript. See Information for Authors for further details.
- You must start the Methods section with a paragraph headed Ethical Approval. A detailed explanation of journal policy and regulations on animal experimentation is given in Principles and standards for reporting animal experiments in The Journal of Physiology and Experimental Physiology by David Grundy J Physiol, 593: 2547-2549. doi:10.1113/JP270818). A checklist outlining these requirements and detailing the information that must be provided in the paper can be found at: <https://physoc.onlinelibrary.wiley.com/hub/animal-experiments>. Authors should confirm in their Methods section that their experiments were carried out according to the guidelines laid down by their institution's animal welfare committee, and conform to the principles and regulations as described in the Editorial by Grundy (2015), including an ethics approval reference number. The Methods section must contain a statement about access to food, water and housing, details of the anaesthetic regime: anaesthetic used, dose and route of administration, and method of killing the experimental animals.
- Your manuscript must include a complete Additional Information section, including competing interests; funding; author contributions and acknowledgements.
- Please upload separate high-quality figure files via the submission form.
- Please ensure that the Article File you upload is a Word file.
- Papers must comply with the Statistics Policy: https://jp.msubmit.net/cgi-bin/main.plex?form_type=display_requirements#statistics.

In summary:

- If n {less than or equal to} 30, all data points must be plotted in the figure in a way that reveals their range and distribution. A bar graph with data points overlaid, a box and whisker plot or a violin plot (preferably with data points included) are acceptable formats.

- If $n > 30$, then the entire raw dataset must be made available either as supporting information, or hosted on a not-for-profit repository, e.g. FigShare, with access details provided in the manuscript.
- 'n' clearly defined (e.g. x cells from y slices in z animals) in the Methods. Authors should be mindful of pseudoreplication.
- All relevant 'n' values must be clearly stated in the main text, figures and tables.
- The most appropriate summary statistic (e.g. mean or median and standard deviation) must be used. Standard Error of the Mean (SEM) alone is not permitted.
- Exact p values must be stated. Authors must not use 'greater than' or 'less than'. Exact p values must be stated to three significant figures even when 'no statistical significance' is claimed.

- Please include an Abstract Figure file, as well as the Figure Legend text within the main article file. The Abstract Figure is a piece of artwork designed to give readers an immediate understanding of the research and should summarise the main conclusions. If possible, the image should be easily 'readable' from left to right or top to bottom. It should show the physiological relevance of the manuscript so readers can assess the importance and content of its findings. Abstract Figures should not merely recapitulate other figures in the manuscript. Please try to keep the diagram as simple as possible and without superfluous information that may distract from the main conclusion(s). Abstract Figures must be provided by authors no later than the revised manuscript stage and should be uploaded as a separate file during online submission labelled as File Type 'Abstract Figure'. Please also ensure that you include the figure legend in the main article file. All Abstract Figures should be created using BioRender. Authors should use The Journal's premium BioRender account to export high-resolution images. Details on how to use and access the premium account are included as part of this email.

EDITOR COMMENTS

Reviewing Editor:

Your manuscript has been reviewed by two experts in the field. Both provide valuable suggestions to improve the manuscript. Please carefully go through those suggestions and respond in a step by step manner.

While reviewer two has a valid point that many experiments are these days performed in awake animals using large scale multi-channel recordings, I disagree that all experiments have to be performed that way. Recordings in anesthetized animals have the advantage of being less prone to be influenced by stress levels of the animal and interpretations due to yet unexplainable interactions within the brain. However, in the present manuscript I do see a risk in the choice of anesthesia. Isoflurane is known to inhibit Ih channels and as one of the findings in the present manuscript reports Ih independent generation of rebound firing - choosing isoflurane presents a conflict. Adding a proof of concept that the results also holds when Ih is not blocked would support the findings.

Please also see 'Required Items' above.

Senior Editor:

We are pleased to report that your manuscript is considered to be of potential interest. However, the reviewers have brought up several points around the analysis of the data that require addressing by the authors. Also, the reviewing editor has raised a good point about a potential confound around isoflurane effects on Ih channels that likely will require some additional experiments. In addition, the authors will need to be clear in the abstract that their studies are being done in anesthetized animals and discuss any potential caveats with the approach along with advantages (e.g., those noted by the reviewing editor). We do not believe that large-scale recordings in awake animals are required for the questions addressed here, but additional attention to this point is needed.

REFEREE COMMENTS

Referee #1:

Shapiro et al. characterize rebound activity following optogenetic photostimulation of inhibitory interneuron classes in anesthetized mouse V1. The manuscript addresses this interesting and oft ignored issue systematically and extensively. The authors show that putative pyramidal neurons show rebound activity following mild optogenetic drive of Pvalb, SOM and VIP

interneurons, albeit to different degrees and with a large cell-to-cell variation. Pvalb-stimulation leads to the most marked rebounds in pyramidal neurons. Interestingly, rebound magnitudes are positively correlated with the visual afferent drive (elicited by grating stimuli of different contrasts). This is surprising due to the expectation that rebounds are mostly driven by hyperpolarization-activated Ih currents, which would lead to the opposite prediction. The authors further investigate post-photostimulation effects in the interneurons themselves, showing a faster activity decay especially in Pvalb+ neurons, if visual stimulation offset is coupled to optogenetic stimulation offset. Using local bar stimuli, it is shown that visual drive dependency of rebounds is locally restricted. Finally, by temporally uncoupling visual and optogenetic stimulation the authors show interactivity of visual drive with rebound.

The manuscript is well written and experiments as well as analyses are mostly to the point. The authors provide a valuable addition to our understanding of inhibitory/excitatory circuit interactions in the neocortex, by systematically investigating the effects across different interneuron classes and comparing their effects on pyramidal neurons across different visual drive levels. The authors show the surprising phenomenon that rebounds are stronger in high visual drive conditions and provide a mechanistic explanation that is not pyramidal-neuron cell-intrinsic (Ih) dependent but based on amplified after-hyperpolarization in interneurons and subsequent network effects. The overall findings and conclusions are well-founded in the reported evidence.

In the details, I see a few issues, which, if addressed, would strengthen the paper.

* The main distinction between cell groups is the reported photomodulation latency, with the apparent assignment as pyramidal neurons if the latency was above 30 ms. No further attempts at distinguishing inhibitory and pyramidal neurons (e.g. regular vs. fast spiking) is reported. Would the inclusion of various cell types, also interneurons explain some of the cell-by-cell variability? Similarly, can some of the variation be attributed to the lateral and layer distance of the pyramidal neurons to the stimulation focus?

* I do not see the value in the "quartile analysis" shown in Figs. 4 and 6. The main conclusion hereby seems circular to me: if we sort the neurons by rebound, the highest neurons have the strongest rebound. Time varying signals during the photostimulated period are highly variable within quartiles and thus their average is not really informative. Especially in the center three quarters of the populations the sorting seems to be driven by minute differences. I would prefer either some sort of clustering approach or scatter plots comparing the photostimulated period to the rebound phase. The relationship of the Z-score function during photostimulation and "neg. rebound" (Fig. 6) could be interesting, but also here the quartile analysis does not help to explain cell-by-cell variability. For example, would the decay function during stimulation at 100% contrast be predictive of post-stimulation decay?

* In the analyses of the post-photostimulation effects in the stimulated interneurons themselves, I think the terminology of "negative rebound" is misleading. Obviously, spike rates cannot be negative, and the authors defined rebound as the difference between photostimulated and not photostimulated conditions over identical visual stimulus conditions, which is practicable and viable for pyramidal neurons. Still, the term rebound implies a change over baseline and less of a difference in response and thus I find that a term like "fast decay" would be more appropriate for the post-photostimulated effects in stimulated interneurons.

* In Fig. 8, Z-score functions and especially rebounds seem much smaller than during Experiment 1 (8B vs. 3A), which needs an explanation. The asserted presence of more complex and prolonged rebounds in the HiHi condition (8C), especially compared with Fig. 3A seems not to be supported by the data shown. Showing heatmaps of individual cell responses as in Figs. 4 & 6 might help to underline this point.

* It should be discussed how isoflurane anesthesia could affect excitatory-inhibitory balance and dynamics in respect to rebounds.

* The relationship between optogenetically induced rebounds and potentially similar physiologically evoked events should be discussed

Minor:

I find the contrast-level colormap (Figs. 2, 3, 5) not intuitive and would prefer a more linear colormap

Referee #2:

The work of Shapiro et al. examines the off response of the cortical network (in mouse V1) to an optogenetic stimulus targeted to PV+, SOM+ and VIP+ inhibitory cells. The question itself is of interest, however to my mind the experimental and analytical level of the work is below the standard one typically expects from JPhysiol. publications.

1. All the experiments are carried out under anesthesia. This should be mentioned in the abstract, as this is no longer the standard in the field. In particular, this raises the question of whether the findings will carry over to awake cortex, or depend on the anesthetized brain state. Indeed, the lack of background spontaneous activity is important for the key observations and is made up for by strong visual stimuli, yet this almost surely is the result of the animals being under deep anesthesia.
2. The recordings are performed with a single tetrode, which is a rather outdated method. More importantly, no information is provided on how spike sorting quality was assessed, so it is not clear if most of the "neurons" in the manuscript are indeed properly well-isolated spike clusters.
3. To the extent that fig. 2 shows the typical best examples, the visual responses raise questions: for a drifting grating at 50-100% contrast we expect to see a clear oscillatory PSTH, which is the case in panel B, but not A or C.
4. Throughout the ms, we are only shown what happens to the right of time 0 (stimulus onset). Yet, it is interesting to see the baseline as well, e.g., during the 500ms before time 0. Indeed, it seems that at baseline neurons are (mostly?) silent, which might be due to deep anesthesia (and very much unlike the awake brain, as already mentioned in #1)
5. The main analysis used throughout the ms obfuscates key information. The work compares visual responses with and without optogenetic interventions. Instead of showing the population average PSTH (perhaps both the literal average and with normalization, such that baseline activity is 1 for each neuron, i.e. neurons are given the same weight irrespective of their firing rate) in each condition, all we are shown is the population average of the ratio (so called z-score). This is insufficient: $10/1 = 1/0.1 = 10$, yet the interpretation of 10/1 and 1/0.1 is very different. For example, in fig. 7 we are shown that flashed bar causes a rebound at population level, yet whether this is because there is no off response without optogenetic stimulus (as in the one example neuron) or the optogenetics makes this off response somewhat more prominent (as I suspect) is not shown.

Similarly, Fig. 4A shows that optogenetics on its own (e.g. with 6% contrast) has no effect, which is strange and surprising - it should suppress spontaneous spiking of Pyr cells. For 100% contrast, there's strong suppression, which at first sight appears as strong visual stimulus producing suppression (very counterintuitive), whereas this is actually the result of showing the z-scored ratio.

As an aside, if the recordings would have been performed with modern high-channel silicon probes (#2 above), population PSTH could have been computed from each recording, and not just by pooling all the recordings together.

6. On the positive side, the text of the ms is for the most part well and clearly written, and easy to follow.

END OF COMMENTS

Senior and Reviewing Editors

Comment 1: Regarding anesthesia, the Reviewing Editor wrote: “[While] many experiments are these days performed in awake animals using large scale multi-channel recordings, I disagree that all experiments have to be performed that way.” However, they also astutely noted that our explanation of rebound activity focusing on I_h channels was problematic because the isoflurane anesthesia we used can interfere with I_h function. In a clarifying email, the advice of the Senior and Reviewing Editors was: “Additional proof of concept experiments with another anaesthetic are not needed and a discussion of the caveats associated with the use of isoflurane is sufficient.”

Response 1: We thank the Editors for their initial insights, as well as for taking the time to clarify their advice to us. We have modified the manuscript’s title, and made clear in the Abstract (pg. 3 [line 58]) and Introduction (pg. 6 [lines 146-147]) that our experiments were done in anesthetized animals. Importantly, we have added a paragraph in the Discussion (pg. 47-48 [lines 1124-1162]) explaining the benefits and potential caveats of measuring neural activity in anesthetized animals, with particular consideration of isoflurane.

Comment 2: “Perhaps the authors could discuss Bruce Bean's nice recent JP review on the whether or not I_h contributes to rebounds and rhythms.”

Response 2: We thank the Editors for this extremely helpful advice. We found the Bean (2024) review to be highly relevant, and it opened our eyes to several additional mechanisms that can produce pacemaking and rebounds. We have made major overhauls to the Introduction (pg. 5-6 [lines 112-117; 131-138; 156-159]) and Discussion (pg. 43-44 [lines 999-1023]) to present a broader consideration of cellular rebound mechanisms. We feel that moving away from the exclusive focus on I_h was a major improvement for the manuscript.

Comment 3: “We do not believe that large-scale recordings in awake animals are required for the questions addressed here”.

Response 3: We appreciate the Editor’s openness to a variety of technical approaches.

Reviewer 1

Comment 1: “The main distinction between cell groups is the reported photomodulation latency...No further attempts at distinguishing inhibitory and pyramidal neurons (e.g. regular vs. fast spiking) is reported. Would the inclusion of various cell types, also interneurons explain some of the cell-by-cell variability?”

Response 1: To clarify, we used both photomodulation latency and sign (suppression vs. facilitation; pg. 13 [line 346]) to distinguish photostimulated interneurons because logically only channelrhodopsin2-expressing interneurons could be activated at such low latencies. Nevertheless, we agree with the reviewer that under this classification scheme our putative Pyramidal neurons (Pyr) category could be heterogeneous and contain some narrow spiking cells, which are often associated with interneurons. We added a short analysis of spike waveform (peak-to-trough interval) to Experiment 1 where we were examining between-cell variability, but didn’t find any significant relationships with

rebound magnitude (pg. 21-22 [lines 513-521]). We elected not to delve too deeply into spike waveform differences because many Pyr neurons (especially in infragranular layers) also have narrow spikes (Dykes et al. 1988; Swadlow 1988, 1990, 1991, 1994, 2003; Zhuang et al. 2013), and even the parvalbumin expressing interneurons (Pvalb+) that have been most closely associated with narrow-spiking waveforms are not always narrow-spiking (Blatow et al. 2003; Senzai et al. 2019).

Comment 2: “Similarly, can some of the variation be attributed to the lateral and layer distance of the pyramidal neurons to the stimulation focus?”

Response 2: To address layer differences as a potential explanation for between-cell variation, we have added scatter plots comparing rebound magnitude with tetrode recording depth for both Pyr cells (Figure 4 C,F,I,L; pg. 21 [lines 509-513]) and interneurons (Figure 6 C,F,I; pg. 28 [lines 638-647]). We did not analyze lateral distance between the recording site and illumination because the 0.4 mm diameter fiberoptic was in a fixed position relative to the tetrode shaft and provided uniform illumination meaning that the lateral distance from the light did not change between recording sites.

Comment 3: “I do not see the value in the "quartile analysis" shown in Figs. 4 and 6... I would prefer either some sort of clustering approach or scatter plots comparing the photostimulated period to the rebound phase.”

Response 3: As suggested, we have deleted the quartile analyses from both figures 4 and 6, and replaced them with scatter graphs comparing mean Z-scores in the photostimulation vs. the post-photostimulation epochs (Pyr cells: Figure 4B,E,H,K described on pg. 21 [lines 503-509]; Interneurons: Figure 6B,E,H described on pg. 28 [lines 629-637]). We agree with the reviewer that this analysis, along with the consideration of layer differences outlined in Comment 2 above provide a more useful examination of between-cell variation in rebound magnitude.

Comment 4: “In the analyses of the post-photostimulation effects in the stimulated interneurons themselves, I think the terminology of "negative rebound" is misleading... the term rebound implies a change over baseline and less of a difference in response and thus I find that a term like "fast decay" would be more appropriate for the post-photostimulated effects in stimulated interneurons.”

Response 4: We understand the reviewer’s perspective. To avoid confusion, we have replaced the confusing “negative rebound (neg-rebound)” with the more descriptive “Post-Excitation Rapid Decay (PERD)” throughout the manuscript.

Comment 5: “In Fig. 8, Z-score functions and especially rebounds seem much smaller than during Experiment 1 (8B vs. 3A), which needs an explanation.”

Response 5: We thank the reviewer for alerting us to this difference. We believe the lower rebound magnitude in Experiment 3 might be partially explained by the shorter photostimulation time we had to use (1000ms in Experiment 1 vs. 500ms in Experiment 3). Li et al. (2019) also showed that going from 1000ms to 500ms photostimulation duration approximately halved their rebound magnitude. We have now explicitly pointed out this difference in the Results section (pg. 37 [lines 828-832]), and noted the similarity to Li et al. (2019).

Comment 6: “The asserted presence of more complex and prolonged rebounds in the HiHi condition (8C), especially compared with Fig. 3A seems not to be supported by the data shown. Showing heatmaps of individual cell responses as in Figs. 4 & 6 might help to underline this point.”

Response 6: We have added population heatmaps to Experiment 3 for Pyr cells (figure 8G) and Pvalb+ interneurons (figure 9J). We believe the reviewer is correct, this does help illustrate that the rebounds in the HiHi condition were prolonged and sometimes vacillated between positive and negative Z-scores. These heatmaps also enabled us to provide more detailed descriptions of rebounds (pg. 37 [lines 824-840]) and PERD (pg. 40 [lines 906-912]) for each condition (HiLo, HiHi, LoHi, and LoLo).

Comment 7: “It should be discussed how isoflurane anesthesia could affect excitatory-inhibitory balance and dynamics in respect to rebounds.”

Response 7: This comment was also raised by the Senior and Reviewing Editors. We have added a paragraph in the Discussion (pg. 47-48 [lines 1124-1162]) describing the potential influence of isoflurane anesthesia on the rebound effects we observed.

Comment 8: “The relationship between optogenetically induced rebounds and potentially similar physiologically evoked events should be discussed.”

Response 8: We have added a Conclusions paragraph to our Discussion (pg. 48-49 [lines 1164-1181]) where we speculate on the role of potential cellular and network rebound mechanisms during physiologically evoked events. We propose the multiple mechanisms that could contribute to optogenetic-mediate rebounds would normally regulate cellular and/or network excitation and inhibition within a narrow operating range.

Comment 9: “I find the contrast-level colormap (Figs. 2, 3, 5) not intuitive and would prefer a more linear colormap”

Response 9: We have changed the contrast-level color scheme for figures 2, 3, and 5 to a linear black-to-pink gradient, which we hope is more intuitive. We were pleased that this suggestion also made these figures easier to interpret when printed in monochrome.

Reviewer 2

Comment 1: “All the experiments are carried out under anesthesia. This should be mentioned in the abstract, as this is no longer the standard in the field.”

Response 1: We have modified the manuscript’s title, and made clear in the Abstract (pg. 3 [line 58]) and Introduction (pg. 6 [lines 146-147]) that our experiments were done in anesthetized animals.

Comment 2: “...this raises the question of whether the findings will carry over to awake cortex, or depend on the anesthetized brain state. Indeed, the lack of background spontaneous activity is important for the key observations and is made up for by strong visual stimuli, yet this almost surely is the result of the animals being under deep anesthesia.”

Response 2: We appreciate the reviewer's scepticism, and there are several issues to unpack with this comment. First, we cite papers that report the basic phenomenon of optogenetic-mediated rebounds in awake mice (e.g. Stark et al., 2013; Li et al., 2019), anesthetized mice (e.g. Chuong et al., 2014; Assaf and Schiller, 2016), and even brain slices (e.g. Madisen et al. 2012; Bohannon and Hablitz, 2018), so it is clear that these types of rebounds do not depend on anesthesia to be observed. Alternatively, the reviewer may be referring to our findings that rebounds were larger following strong afferent drive, but to this point we view the ability to control network activity with visual stimuli as an advantage of recording under anesthesia. We have added a paragraph about anesthesia to the Discussion (pg. 47-48 [lines 1124-1162]) where we note that the spontaneous firing and high trial-to-trial variability in awake mice have largely been attributed to cortex-wide signals related to spontaneous movements, choice behavior on operant tasks, and arousal/attention (e.g. Stringer et al. 2019; Musall et al. 2019; Salkoff et al. 2020). We emphasize that using awake mice would make our experiments more challenging because we would have less control over local network activity. Finally, the assertion that our data was result of the animals being under deep anesthesia ignores the fact that visually evoked activity disappears under higher doses of isoflurane. We clarify in the new Discussion paragraph that vision studies on anesthetized mice (admittedly less popular currently) must carefully titrate the depth of anesthesia to produce unconsciousness but not excessive suppression.

Comment 3: "The recordings are performed with a single tetrode, which is a rather outdated method. More importantly, no information is provided on how spike sorting quality was assessed, so it is not clear if most of the "neurons" in the manuscript are indeed properly well-isolated spike clusters."

Response 3: We have added details to our description of spike-sorting in the Methods section (pg. 9 [lines 228-234]), where we emphasize that only clearly separated clusters from principal component analysis were included in our datasets. It is true that tetrodes trade-off channel count for affordability, but tetrodes share advantages with silicone probes that have closely spaced recording sites in that they improve discriminability when a waveform appears on more than one site (see Niell and Stryker (2008) for a comparison of NeuroNexus probes with 25 μ m vs. 50 μ m site spacing in mouse V1).

Comment 4: "To the extent that fig. 2 shows the typical best examples, the visual responses raise questions: for a drifting grating at 50-100% contrast we expect to see a clear oscillatory PSTH, which is the case in panel B, but not A or C."

Response 4: The reviewer's statement that all neurons in primary visual cortex (V1) are expected to produce phase-sensitive responses to drifting gratings is incorrect. First, considering all cortical layers V1 contains ~50% phase-sensitive cells (sometimes called linear or simple cells), and ~50% phase-insensitive (sometimes called nonlinear or complex cells; see Niell and Stryker (2008) for anesthetized mouse data). Second, phase-sensitive cells will only show nice half-wave rectified oscillatory responses at their preferred spatial frequency (Skottun et al. 1991). We did not tailor the spatial frequency of our stimuli to each neuron (pg. 10 [pg. 257-259]), but that should not affect our findings regarding rebounds. We selected the example cells to show clear monotonic

increases in visually evoked responses to contrast as well as clear rebounds. We do not see a problem with showing a mix of phase-sensitive and phase-insensitive examples.

Comment 5: “Throughout the ms, we are only shown what happens to the right of time 0 (stimulus onset). Yet, it is interesting to see the baseline as well, e.g., during the 500ms before time 0. Indeed, it seems that at baseline neurons are (mostly?) silent, which might be due to deep anesthesia (and very much unlike the awake brain, as already mentioned in #1).”

Response 5: We find this comment puzzling because the baseline firing rate is shown for example neurons throughout the manuscript. First, the example neurons in fig. 8A, and 9A-D show exactly what the reviewer describes – 500ms of neural activity before stimulus onset. Second, for the example neurons in figures 2A-C and 5A spiking is shown returning to baseline after around 1300ms. Individual and population Z-score functions are also shown returning to 0 (where there is no difference in spontaneous firing between the control and photostimulated conditions) in figures 2-6, 8, and 9. Nevertheless, we imagine that some readers could prefer to see longer periods of baseline activity, so we have also added normalized population responses in figure 2D-G, which now show mean baseline firing for an even longer duration.

Once again, we acknowledge that baseline firing will be lower in anesthetized animals compared to awake behaving mice, but repeat that measuring visually evoked activity as we have done would not be possible under excessively deep anesthesia. All the example cells showing spike rates indicate that spontaneous firing is generally between 0.1-10 spikes/s, which is similar to what has been reported by other V1 studies in anesthetized mice (e.g. Niell and Stryker, 2008).

Comment 6: “The main analysis used throughout the ms obfuscates key information. The work compares visual responses with and without optogenetic interventions. Instead of showing the population average PSTH (perhaps both the literal average and with normalization, such that baseline activity is 1 for each neuron, i.e. neurons are given the same weight irrespective of their firing rate) in each condition, all we are shown is the population average of the ratio (so called z-score). This is insufficient: $10/1 = 1/0.1 = 10$, yet the interpretation of $10/1$ and $1/0.1$ is very different.”

Response 6: There are several issues to unpack with this comment.

First, the Z-score functions that we used are more like a normalized difference ($Z = [\varphi - \mu] / \sigma$; see pg. 12) rather than a ratio. In this calculation, μ will normally be close to 0 since the spontaneous firing for the control and photostimulated conditions will be similar, so Z really reflects the difference in firing during the photostimulation and post-photostimulation epochs (φ) normalized by the variability of the spontaneous firing (σ). We have clarified our motivation for using Z-score functions to standardize the magnitude of optogenetic effects across neurons in the Methods section (pg. 12 [lines 317-322]). We agree with the reviewer’s point about interpreting ratios, but this is not relevant to the calculation we used.

Second, we agree with the reviewer that some depiction of rebounds at the population level before the Z-scores are calculated could help readers conceptualize the data we were working with. Thus, we have included new panels in figure 2D-G showing the normalized population responses to 100% gratings in the control and photostimulated conditions for each dataset. However, we feel it would be overly repetitive to show this

type of data for all conditions throughout the manuscript (nor does J. Physiol. advise using supplemental figures). This is because it is not enough to show PSTHs, an actual subtraction (as in our Z-score metric) must be done to quantify the differences between conditions. These Z-scores allow us to summarize the effects in Experiments 1-3 in a more compact way and also enable us to use heatmaps to show variability within the datasets. The Z-score panels above each spike density function for the example neurons in figures 1, 2 and 5 (and heatmap in figure 7C) explicitly show how differences in spike rates between control and photostimulated conditions translate into Z-score functions.

Finally, we would like to emphasize that the Z-score calculation is not required to observe our results. Rebounds are evident with no normalization at all (as in the example cells in Figs. 2A-C, 7A, 8A), whether responses were normalized by baseline firing or max firing (Figs. 2D-G), or using Z-scores (Figs. 3-9). We believe normalizing using the Z-score calculation is the most sensible because it takes variability in spiking into account when measuring the difference between photostimulation and control conditions. For example, a relatively small increase in firing in the post-photostimulation epoch (compared to the large visually evoked responses) may not be important if the baseline firing is highly variable, but is more likely to be important if the baseline firing has low variability.

Comment 7: “Similarly, Fig. 4A shows that optogenetics on its own (e.g. with 6% contrast) has no effect, which is strange and surprising - it should suppress spontaneous spiking of Pyr cells.”

Response 7: We clearly show that interneuron photostimulation does produce suppression for 6% contrast gratings. Figure 3A,C,E shows pyramidal cell datasets where the population average Z-scores for 6% contrast were below 0 during the photostimulation epoch (Fig. 3A shows the same dataset as Fig. 4A, but in a different format). Importantly, it is totally expected that the suppression at low contrasts will be smaller than the suppression at high contrasts, whether the data is shown as spike rates or Z-scores. First, due to the spike threshold nonlinearity spike rates cannot go below zero. Second, photostimulation of interneurons often produces divisive inhibition in pyramidal cells with proportionally less suppression at lower spike rates (Wilson et al., 2012; El-Boustani and Sur, 2014; El-Boustani et al., 2014; Ingram et al. 2019), which is thought to scale or normalize the responses of neurons while maintaining their selectivity (Carandini and Heeger, 1994; Salinas and Thier, 2000).

Comment 8: “[Referring to Fig. 4A...] For 100% contrast, there's strong suppression, which at first sight appears as strong visual stimulus producing suppression (very counterintuitive), whereas this is actually the result of showing the z-scored ratio.”

Response 8: We feel that the examples cells shown in Figures 1 and 2 should provide the reader with an understanding that negative Z-scores indicate suppression. The scale bar in figure 4 is also clearly labeled to indicate the heatmaps depict differences between control and photostimulated conditions as Z-scores. From a practical perspective, using Z-scores in figure 4 (or some similar difference calculation) is the only way to show this density of data that allows comparisons across cells and stimulus conditions.

Dear Dr Crowder,

Re: JP-RP-2025-287265R1 "Characterizing Optogenetic-mediated Rebound Effects in Anesthetized Mouse Primary Visual Cortex" by Jared T Shapiro, Nicole M Michaud, and Nathan A Crowder

Thank you for submitting your manuscript to The Journal of Physiology. It has been assessed by a Reviewing Editor and by 2 expert referees and we are pleased to tell you that it is acceptable for publication following satisfactory revision.

REVISION CHECKLIST:

We look forward to receiving your revised submission.

Yours sincerely,

Nathan Schoppa
Senior Editor
The Journal of Physiology

REQUIRED ITEMS

- You must start the Methods section with a paragraph headed Ethical approval (https://jp.msubmit.net/cgi-bin/main.plex?form_type=display_requirements#methods).

Research must comply with The Journal's policies regarding animal experiments (<https://physoc.onlinelibrary.wiley.com/hub/animal-experiments>) and adherence to these policies must be stated in the manuscript.

Authors should confirm in their Methods section that their experiments were carried out according to the guidelines laid down by their institution's animal welfare committee, including an ethics approval reference number. The Methods section must contain a statement about access to food, water and housing, details of the anaesthetic regime: anaesthetic used, dose and route of administration, and method of killing the experimental animals.

EDITOR COMMENTS

Reviewing Editor:

Thank you for your thoughtful revisions. There are a few more minor changes suggested by reviewer 1.

Please also include information of the use of analgesia in your experiments (in addition to the anesthesia = isoflurane and relaxant = chlorprothixene).

Senior Editor:

Thank you for submitting your thoroughly revised manuscript. The paper has been re-evaluated by the two original referees and, while their opinions varied somewhat, we believe that overall the authors have done an excellent job of addressing the prior concerns and that the work will make a substantial contribution to the field. The authors should address the remaining minor concerns from Referee 1. I will echo Referee 1's point that the mainly speculative discussion of mechanisms is very long (sometimes with very long paragraphs), and these sections should be significantly shortened. At the same time, to address one of the concerns of Referee 2, around the use of the z-scores, the authors should add a brief discussion highlighting that their basic result was seen not just with z-scores, but also completely non-normalized data and data normalized to the background firing rates. The authors' point is now well made in the last paragraph of their rebuttal to Referee 2's Comment 6. Also, if this is not already included in their discussion of the use of isoflurane anesthesia in the Discussion, the authors should discuss how the low spontaneous firing rate that is observed under anesthesia may have affected their results. I see a discussion now about mechanisms directly affected by isoflurane but not necessarily how isoflurane's effects on spontaneous firing may have altered their analysis.

REFEREE COMMENTS

Referee #1:

The authors addressed all of the points raised in my previous review. In particular the addition of scatter plots illustrating the relationship between photostimulation effect and rebound/PERD while accounting for cell-by-cell variability help to clarify this major point. The newly added points in the discussion are important and give nuance to the interpretations.

The paper gives a convincing account that photostimulation rebounds can depend on sensory drive in a cell-type and connectivity-dependent manner. As such it is an important notification for other groups using photostimulation of interneurons to check for the presence of rebounds at different activity levels and timings.

Concerning the mechanistic explanations the paper suggests potential cell-intrinsic and network effects, however the evidence here is weaker. The authors chose relatively mild photostimulation intensities, where rebound effects are also relatively mild and only substantially appear at high sensory drive. While underlining the potential of accidentally overlooking rebound effects in optogenetic experiments, the weak and diverse effects complicate mechanistic conclusions. Varying photostimulation intensity and temporal patterns (e.g. different durations, pulse strobing), could have helped disentangling cell-intrinsic effects in the stimulated cells from network effects. While I agree with the authors and editors that awake recordings are not necessary for the research questions raised here and might even make interpretations harder, any of the physiological mechanisms could be affected by the anesthetic of choice. So we do simply not know if rebounds in this experimental conditions are smaller or larger with different/no anesthesia. Nevertheless, the presented data indicates that rebound mechanisms are probably diverse and interact on multiple levels, which is interesting in its own right.

Minor

* The discussion section appears slightly bloated and could benefit from some textual restructuring, better emphasizing the major take-aways of the manuscript and de-emphasizing mechanistic musings.

* I'd prefer more expressive titles for the three experiments (e.g. sensory intensity effects, spatial effects, temporal effects) to focus on the raised questions

* Figure 5H is discussed after Figure 6 in result text.

* in Fig 3BDFH, 5CEG it is not explicit in figure or legend that we are looking at the rebound/PERD phase. It is mentioned in the result text but should be explicit here.

Referee #2:

I would like to thank the authors for addressing the comments that were raised. The new version was clearly improved, however since the edits comprise a moderate level of alterations to the original story, and my comments were addressed only partially, my overall assessment of the work did not change.

1. Addressed to a large extent, thanks. I disagree with the assertion made in the rebuttal that "visually evoked activity disappears under higher doses of isoflurane".

2. Addressed, but the method is incorrect. First, 1ms is too low: 2 or 3ms should have been used. Second, using 1% threshold is incorrect, because the threshold should depend

on the firing rate of the neuron (briefly: for a 10 spk/s neuron 1% is ok, but for a 0.1 spk/s neuron, not at all; e.g. see Hill, Mehta, Kleinfeld, JNeurosci. 2011)

3. I did not claim that all neurons in V1 show oscillatory PSTH. If the authors believe that these 3 neurons are representative for their dataset, it's their choice. But as a reader, it does not add to my confidence in the results.

4. Not addressed. I agree that figs. 8 and 9 indeed show pre-stimulus baseline, so my original phrasing "throughout the ms" was inaccurate, but the point that this is the case throughout most of the ms, and that showing pre-stimulus baseline would be worthwhile, was presumably clear.

5. Thank you for including fig. 2D-G, I believe they add to the story.

I agree that showing this throughout along with Z-scored analysis is not practical. My opinion about using Z-score as the main quantification did not change, which was the reason for my comment about what fig. 4A shows (which still stands), but I see no point in belaboring the same issue again and again.

END OF COMMENTS

Reviewing Editor Comments

Comment 1: “Thank you for your thoughtful revisions. There are a few more minor changes suggested by reviewer 1.”

Response 1: As requested, we have addressed all the minor changes suggested by reviewer 1. Point-by-point details are provided below.

Comment 2: “Please also include information of the use of analgesia in your experiments (in addition to the anesthesia = isoflurane and relaxant = chlorprothixene).”

Response 2: We did not use additional analgesia because our electrophysiology experiments were terminal (animals were anesthetized at the start and euthanized without ever gaining consciousness), and no neuromuscular blocking agents were used. We have now stated this explicitly on pg. 8-9 (lines 221-223). We have also added a few more details on anesthetic depth to ensure readers know we have taken all steps to minimise the animals’ pain or distress (pg. 8 [lines 217-220]). We take animal welfare extremely seriously, and note that the approval of our experimental protocol provided by our local ethics committee aligns perfectly with the guidelines of the Journal of Physiology.

Senior Editor Comments

Comment 1: “Thank you for submitting your thoroughly revised manuscript. The paper has been re-evaluated by the two original referees and, while their opinions varied somewhat, we believe that overall the authors have done an excellent job of addressing the prior concerns and that the work will make a substantial contribution to the field. The authors should address the remaining minor concerns from Referee 1.”

Response 1: We thank the editors and reviewers for their thoughtful feedback. As requested, we have addressed all the minor changes suggested by reviewer 1. Point-by-point details are provided below.

Comment 2: “I will echo Referee 1's point that the mainly speculative discussion of mechanisms is very long (sometimes with very long paragraphs), and these sections should be significantly shortened.”

Response 2: As requested, we have shortened the Discussion sections titled “*Cellular Rebound Mechanisms*” and “*Network Rebound Mechanisms*” from 1906 to 1143 words, which is a reduction of 40%. We agree that streamlining these sections (pg. 43-46 [lines 998-1086]) will help emphasize the main findings and make the manuscript more digestible.

Comment 3: “At the same time, to address one of the concerns of Referee 2, around the use of the z-scores, the authors should add a brief discussion highlighting that their basic result was seen not just with z-scores, but also completely non-normalized data and data normalized to the background firing rates. The authors' point is now well made in the last paragraph of their rebuttal to Referee 2's Comment 6.”

Response 3: We agree that this important point should appear in the manuscript. We have added a short paragraph to the discussion (pg. 42-43 [lines 962-972]) indicating our basic results could be observed using various analytical methods.

Comment 4: “Also, if this is not already included in their discussion of the use of isoflurane anesthesia in the Discussion, the authors should discuss how the low spontaneous firing rate that is observed under anesthesia may have affected their results. I see a discussion now about mechanisms directly affected by isoflurane but not necessarily how isoflurane's effects on spontaneous firing may have altered their analysis.”

Response 4: We have added text to the Discussion to clarify that the low spontaneous firing rate observed under anesthesia likely helped us detect rebounds with mild photostimulation and demonstrate the change in rebound magnitude with sensory drive (pg. 47 [lines 1122-1124]).

Reviewer #1 comments

Comment 1: “The discussion section appears slightly bloated and could benefit from some textual restructuring, better emphasizing the major take-aways of the manuscript and de-emphasizing mechanistic musings.”

Response 1: This point was echoed by the Senior editor. We have re-worked the Discussion to shorten the “*Cellular Rebound Mechanisms*” and “*Network Rebound Mechanisms*” sections by 40% (pg. 43-46 [lines 998-1086]). We agree that these edits improve the Discussion section substantially by shifting the emphasis onto our findings.

Comment 2: “I'd prefer more expressive titles for the three experiments (e.g. sensory intensity effects, spatial effects, temporal effects) to focus on the raised questions”

Response 2: As requested, we have given our experiments the more descriptive titles of “*Experiment 1: contrast effects*”, “*Experiment 2: spatial effects*”, and “*Experiment 3: timing effects*” throughout the manuscript.

Comment 3: “Figure 5H is discussed after Figure 6 in result text.”

Response 3: We have relocated the paragraph explaining Figure 5H to pg. 28 (lines 623-640) so that it now precedes the description of Figure 6.

Comment 4: “in Fig 3BDFH, 5CEG it is not explicit in figure or legend that we are looking at the rebound/PERD phase. It is mentioned in the result text but should be explicit here.”

Response 4: Thank you for alerting us to this ambiguity. We have edited the caption of Figure 3 to indicate B,D,F,H show the time-averaged rebound Z-scores, and changed their Y-axis labels to indicate they are showing mean rebound Z-scores. Likewise, we have edited the caption of Figure 5 to indicate C,E,G show the time-averaged PERD Z-scores, and changed their Y-axis labels to indicate they are showing mean PERD Z-scores.

Dear Professor Crowder,

Re: JP-RP-2025-287265R2 "Characterizing Optogenetic-mediated Rebound Effects in Anesthetized Mouse Primary Visual Cortex" by Jared T Shapiro, Nicole M Michaud, and Nathan A Crowder

We are pleased to tell you that your paper has been accepted for publication in The Journal of Physiology.

Yours sincerely,

Nathan Schoppa
Senior Editor
The Journal of Physiology

If you would like to receive our 'Research Roundup', a monthly newsletter highlighting the cutting-edge research published in The Physiological Society's family of journals (The Journal of Physiology, Experimental Physiology, Physiological Reports, The Journal of Nutritional Physiology and The Journal of Precision Medicine: Health and Disease), please click this link, fill in your name and email address and select 'Research Roundup':

<https://www.physoc.org/journals-and-media/membernews>

- **TRANSPARENT PEER REVIEW POLICY:** To improve the transparency of its peer review process, The Journal of Physiology publishes online as supporting information the peer review history of all articles accepted for publication. Readers will have access to decision letters, including Editors' comments and referee reports, for each version of the manuscript as well as any author responses to peer review comments. Referees can decide whether or not they wish to be named on the peer review history document.
- You can help your research get the attention it deserves! Check out Wiley's free Promotion Guide for best-practice recommendations for promoting your work at: www.wileyauthors.com/eeo/guide. You can learn more about Wiley Editing Services which offers professional video, design, and writing services to create shareable video abstracts, infographics, conference posters, lay summaries, and research news stories for your research at: www.wileyauthors.com/eeo/promotion.
- **IMPORTANT NOTICE ABOUT OPEN ACCESS:** To assist authors whose funding agencies mandate public access to published research findings sooner than 12 months after publication, The Journal of Physiology allows authors to pay an Open Access (OA) fee to have their papers made freely available immediately on publication.

EDITOR COMMENTS

Reviewing Editor:

Thank you for your careful revisions. Your manuscript can now go into the publishing process.

Senior Editor:

Congratulations! The authors have done a nice job of addressing the prior concerns, and the manuscript is now acceptable for publication.